



# Global sea-level budget and ocean-mass budget, with focus on advanced data products and uncertainty characterisation

**Authors**

Martin Horwath (1), Benjamin D. Gutknecht (1), Anny Cazenave (2,3), Hindumathi Kulaiappan Palanisamy (2, 4), Florence Marti (2), Ben Marzeion (5), Frank Paul (6), Raymond Le Bris (6), Anna E. Hogg (7), Inès Otosaka (8), Andrew Shepherd (8), Petra Döll (9,10), Denise Cáceres (9), Hannes Müller Schmied (9, 10), Johnny A. Johannessen (11), Jan Even Øie Nilsen (11,12), Roshin P. Raj (11), René Forsberg (13), Louise Sandberg Sørensen (13), Valentina R. Barletta (13), Sebastian B. Simonsen (13), Per Knudsen (13), Ole Baltazar Andersen (13), Heidi Randall (13), Stine K. Rose (13), Christopher J. Merchant (14), Claire R. Macintosh (14), Karina von Schuckmann (15), Kristin Novotny (2), Andreas Groh (2), Marco Restano (16), Jérôme Benveniste (17).

(1) Technische Universität Dresden, Institut für Planetare Geodäsie, Dresden, D;
(2) LEGOS Toulouse, F;
(3) International Space Science Institute, Bern, Switzerland
(4) Centre for Climate Research Singapore, Meteorological Service Singapore, Singapore
(5) Institut of Geography and MARUM - Center for Marine Environmental Sciences, University of Bremen, Bremen, Germany
(6) University of Zurich, CH;
(7) University of Leeds, UK;
(8) Centre for Polar Observation and Modelling, University of Leeds, UK;
(9) Institute of Physical Geography, Goethe University Frankfurt, Frankfurt am Main, D;
(10) Senckenberg Leibniz Biodiversity and Climate Research Centre (SBiK-F), Frankfurt am Main, D
(11) Nansen Environmental and Remote Sensing Center, Bergen, NO;
(12) Institute of Marine Research, Bergen, NO.
(13) Technical University of Denmark, DK;
(14) University of Reading and National Centre for Earth Observation, UK;
(15) Mercator Ocean International, Toulouse, F;
(16) Serco/ESRIN, I
(17) ESA ESRIN, I

**Correspondence**

Martin Horwath, martin.horwath@tu-dresden.de





## Abstract

Studies of the global sea-level budget (SLB) and the global ocean-mass budget (OMB) are essential to assess the reliability of our knowledge of sea-level change and its contributions. Here we present datasets for times series of the SLB and OMB elements developed in the framework of ESA's Climate Change Initiative. We use these datasets to assess the SLB and the OMB simultaneously, utilising a consistent framework of uncertainty characterisation. The time series, given at monthly sampling, include global mean sea-level (GMSL) anomalies from satellite altimetry; the global mean steric component from Argo drifter data with incorporation of sea surface temperature data; the ocean mass component from Gravity Recovery and Climate Experiment (GRACE) satellite gravimetry; the contribution from global glacier mass changes assessed by a global glacier model; the contribution from Greenland Ice Sheet and Antarctic Ice Sheet mass changes, assessed from satellite radar altimetry and from GRACE; and the contribution from land water storage anomalies assessed by the WaterGAP global hydrological model. Over the period Jan 1993 – Dec 2016 (P1, covered by the satellite altimetry records), the mean rate (linear trend) of GMSL is $3.05 \pm 0.24$ mm yr$^{-1}$. The steric component is $1.15 \pm 0.12$ mm yr$^{-1}$ (38% of the GMSL trend) and the mass component is $1.75 \pm 0.12$ mm yr$^{-1}$ (57%). The mass component includes $0.64 \pm 0.03$ mm yr$^{-1}$ (21% of the GMSL trend) from glaciers outside Greenland and Antarctica, $0.60 \pm 0.04$ mm yr$^{-1}$ (20%) from Greenland, $0.19 \pm 0.04$ mm yr$^{-1}$ (6%) from Antarctica, and $0.32 \pm 0.10$ mm yr$^{-1}$ (10%) from changes of land water storage. In the period Jan 2003 – Aug 2016 (P2, covered by GRACE and the Argo drifter system), GMSL rise is higher than in P1 at $3.64 \pm 0.26$ mm yr$^{-1}$. This is due to an increase of the mass contributions (now about $2.22 \pm 0.15$ mm yr$^{-1}$, 61% of the GMSL trend), with the largest increase contributed from Greenland. The SLB of linear trends is closed for P1 and P2, that is, the GMSL trend agrees with the sum of the steric and mass components within their combined uncertainties. The OMB budget, which can be evaluated only for P2, is also closed, that is, the GRACE-based ocean-mass trend agrees with the sum of assessed mass contributions within uncertainties. Combined uncertainties (1-sigma) of the elements involved in the budgets are between 0.26 and 0.40 mm yr$^{-1}$, about 10% of GMSL rise. Interannual variations that overlie the long-term trends are coherently represented by the elements of the SLB and the OMB. Even at the level of monthly anomalies the budgets are closed within uncertainties, while also indicating possible origins of remaining misclosures.



# 1 Introduction

Sea level is an important indicator of climate change. It integrates effects of changes of several components of the climate system. About 90% of the excess heat in Earth's current radiation imbalance is absorbed by the global ocean (von Schuckmann et al., 2016, 2020; Oppenheimer et al., 2019). About 3% melts ice (Slater et al., 2021), while the remaining heat warms the atmosphere (1-2%) and the land (~5%). Present-day global mean sea-level (GMSL) rise primarily reflects thermal expansion of sea waters (the steric component) and increasing ocean mass due to land ice melt, two processes attributed to anthropogenic global warming (Oppenheimer et al., 2019). Anthropogenic changes in land water storage (LWS) constitute an additional contribution to the change in ocean mass (Wada et al., 2017; Döll et al., 2014), modulated by effects of climate variability and change (Reager et al., 2016; Scanlon et al., 2018).

To assess the accuracy and reliability of our knowledge about sea-level change and its causes, assessments of the sea-level budget (SLB) are indispensable. Closure of the sea-level budget implies that the observed changes of GMSL equal the sum of observed (or otherwise assessed) contributions, namely the effect of ocean-mass change (OMC) and the steric component (e.g. WCRP, 2018). Steric sea-level can be further separated into volume changes through ocean salinity (halosteric) and ocean temperature (thermosteric) effects, from which the latter is known to play a dominant role in contemporary GMSL rise. Closure of the ocean mass budget (OMB) implies that the observed OMC (e.g., from the Gravity Recovery and Climate Experiment, GRACE, Tapley et al., 2019) is equal to assessed changes of water mass (in solid, liquid or gaseous state) outside the ocean, which are dominated by mass changes of land ice (glaciers and ice sheets) and water stored on land as liquid water or snow. Misclosure of these budgets indicates errors in the assessment of some of the components (including effects of undersampling) or contributions from unassessed elements in the budget.

Over the course of its five assessment reports and its recent Special Report on The Ocean and Cryosphere in a Changing Climate (SROCC; IPCC, 2019), the Intergovernmental Panel on Climate Change (IPCC) has documented a significant improvement in our understanding of the sources and impacts of global sea-level rise. Today, the SLB for the period since 1993 is often considered closed within uncertainties (Church et al., 2013, Oppenheimer et al., 2019). Recent studies that reassessed the SLB over different time spans and using different datasets include the studies by Rietbroek et al. (2016), Chambers et al. (2017), Dieng et al. (2017), Chen et al. (2017, 2020), Nerem et al. (2018), Royston et al, (2020) and Vishwakarma et al. (2020). In the context of the Grand Challenge of the World Climate Research Programme (WCRP) entitled "Regional Sea-level and Coastal Impacts", an effort involving the sea-level community worldwide assessed the various datasets used to estimate components of the SLB during the altimetry era (1993 to present) (WCRP, 2018). A large number of available quality datasets were used for each component, from which ensemble means for each component were derived for the budget assessment.

Significant challenges remain. The IPCC SROCC reported the sum of assessed sea-level contributions for the 1993–2015 period (2006–2015 period) to be 2.76 mm yr$^{-1}$ (3.00 mm yr$^{-1}$, respectively), and this was 0.40 mm yr$^{-1}$ smaller (0.58 mm yr$^{-1}$ smaller) than the observed GMSL rise at 3.16 mm yr$^{-1}$



(3.58 mm yr$^{-1}$) (Oppenheimer et al., 2019, Table 4.1). While the misclosure was within the combined
uncertainties of the sum of contributions and the observed GMSL, these uncertainties were large, with
a 90% confidence interval width of 0.74 mm yr$^{-1}$ to 1.1 mm yr$^{-1}$. Determining the LWS contribution to
sea-level is a particular challenge (WCRP, 2018): Hydrological models suggest LWS losses and
therefore a positive contribution from LWS to GMSL rise, while GRACE analyses suggest LWS gains
and therefore a negative GMSL contribution from LWS. Challenges of SLB assessments include the
question of consistency among the various involved datasets and their uncertainty characterisations. For
example, the study by WCRP (2018) assessed each budget element from a large number of available
datasets generated in different frameworks and used ensemble means of these datasets in the budget
assessment.
ESA's Climate Change Initiative (CCI, https://climate.esa.int) offers a consistent framework for the
generation of high-quality and continuous space-based records of Essential Climate Variables (ECVs;
Bojinski et al., 2014). A number of CCI projects has addressed ECVs relevant for the SLB, most
importantly the Sea-level CCI project, the Sea Surface Temperature (SST) CCI project, the Glaciers
CCI project, the Greenland Ice Sheet CCI project and the Antarctic Ice Sheet CCI project.
The Sea-level Budget Closure CCI (SLBC_cci) project conducted from 2017 to 2019 was the first cross-
ECV project within CCI. It assessed and utilised the advanced quality of CCI products for SLB and
OMB analyses. For this purpose, the project also developed new data products based on existing CCI
products and on other data sources. It is specific to SLBC_cci, and complementary to the WCRP
initiative, that SLBC_cci concentrated on datasets generated within CCI or by project members. The
thorough insights into the genesis and uncertainty characteristics of the datasets facilitated progress
towards working in a consistent framework of product specification, uncertainty characterization, and
SLB analysis.
In this paper we present the methodological framework of the SLBC_cci budget assessments (Sect. 2).
We describe the datasets used, including summaries of the methods of their generation and details on
their uncertainty characterisation (Sect. 3). We report and discuss results of our OMB and SLB
assessments (Sect. 4 to 7) address the data availability in Sect. 8 and conclude in Sect. 9 with an outlook
on suggested work in the sequence of this initial CCI cross-ECV study.
The analysis concentrates on two time periods: P1 from Jan 1993 to Dec 2016 (the altimetry era), and
P2 from Jan 2003 to Aug 2016 (the GRACE/Argo era). The start of P1 is guided by the availability of
altimetry data. Its end is guided by the availability of outputs of the WaterGAP Global Hydrological
Model used in this study to compute LWS, due to availability of climate input data at the time of the
study. The start of P2 is guided by the availability of quality GRACE gravity field solutions at the time
of the study and by the implementation of the Argo drifter array. We note, though, that Argo-based steric
assessments are uncertain in the early Argo years 2003–2004. The budgets are assessed for mean rates
of change (linear trends) over P1 and P2 as well as for GMSL and ocean mass anomalies at monthly
resolution. The OMB assessment also addresses the seasonal cycle.





## 2 Methodological framework

### 2.1 Sea-level budget and ocean-mass budget

The SLB (e.g., WCRP, 2018) expresses the time-dependent sea-level change $\Delta SL(t)$ as the sum of its mass component $\Delta SL_{Mass}(t)$ and its steric component $\Delta SL_{Steric}(t)$:

$$\Delta SL(t) = \Delta SL_{Mass}(t) + \Delta SL_{Steric}(t). \qquad (1)$$

The three budget elements are spatial averages over a fixed ocean domain. We consider the global ocean area in a first instance and we discuss restrictions to sub-areas further below.

More specifically, $\Delta SL(t)$ is the geocentric sea-level change from which effects of glacial isostatic adjustment (GIA) were corrected (Tamisea, 2011; WCRP, 2018). Likewise, assessments of $\Delta SL_{Mass}(t)$ include corrections for GIA effects. The small elastic deformations of the ocean bottom (Frederikse et al., 2017, Vishwakarma et al., 2020) are not corrected in $\Delta SL(t)$ in this study (cf. Section 3.8). The steric component $\Delta SL_{Steric}(t)$ arises from the temporal variations of the height of the sea-water columns of a given mass per unit area in response to temporal variations of the temperature and salinity profiles. The mass component $\Delta SL_{Mass}(t)$ is defined as

$$\Delta SL_{Mass} = \frac{1}{A_{Ocean}\,\rho_W}\Delta M_{Ocean} \qquad (2)$$

where $\Delta M_{Ocean}$ is the change of ocean mass within the ocean domain, $A_{Ocean}$ is the surface area of this domain (defined as $361\cdot10^6$ km$^2$), and $\rho_W = 1000$ kg m$^{-3}$ is the density of water. Change of $A_{Ocean}$ is considered negligible over the assessment period. Equivalently, the mass component can be expressed by a spatial average of the geographically dependent change of ocean mass per surface area, $\Delta\kappa_{Ocean}(x,t)$ (with units of kg m$^{-2}$):

$$\Delta SL_{Mass} = \frac{1}{\rho_W}\langle\Delta\kappa_{source}\rangle_{Ocean} \qquad (3)$$

where $\langle\cdot\rangle_{Ocean}$ denotes the spatial averaging over the ocean domain.

The OMB equation reads

$$\Delta M_{Ocean} = -(\Delta M_{Glaciers} + \Delta M_{Greenland} + \Delta M_{Antarctica} + \Delta M_{LWS} + \text{other}), \qquad (4)$$

where $\Delta M_{Glaciers}(t)$, $\Delta M_{Greenland}(t)$, $\Delta M_{Antarctica}(t)$ and $\Delta M_{LWS}(t)$ are the temporal changes in mass of glaciers outside Greenland and Antarctica (where ice caps are also referred to as glaciers), the Greenland Ice Sheet (GrIS) and Greenland peripheral glaciers, the Antarctic Ice Sheet (AIS), and LWS, respectively. Other terms (e.g., atmospheric water content variations) were not considered in this assessment. We express the OMB in terms of sea-level change,

$$\Delta SL_{Mass} = \Delta SL_{Glaciers} + \Delta SL_{Greenland} + \Delta SL_{Antarctica} + \Delta SL_{LWS} + \Delta SL_{other} \qquad (5)$$

by setting

$$\Delta SL_{Source} = -\frac{1}{A_{Ocean}\,\rho_W}\Delta M_{Source}, \qquad (6)$$

where the suffix "source" stands for Glaciers, Greenland, Antarctica, or LWS.



By expressing the mass component as the sum of the individual mass contributions, the SLB can be
expressed as
$$\Delta\text{SL} = (\Delta\text{SL}_{\text{Glaciers}} + \Delta\text{SL}_{\text{Greenland}} + \Delta\text{SL}_{\text{Antarctica}} + \Delta\text{SL}_{\text{LWS}} + \Delta\text{SL}_{\text{other}}) + \Delta\text{SL}_{\text{Steric}}. \quad (7)$$
For each of the budget equations (1), (5) and (7), we refer to the individual terms on both sides of the
equation as budget elements. We define the budget misclosure as the difference 'left-hand side minus
right-hand side'.
Part of this study refers to the SLB over the ocean area between 65°N and 65°S. This choice is made
because both altimetry and the Argo system have reduced coverage and data quality in the polar oceans.
When referring to a non-global ocean domain, the concept of spatial averaging implied in $\Delta\text{SL}$, $\Delta\text{SL}_{\text{Steric}}$
and $\Delta\text{SL}_{\text{Mass}}$ still holds. However, in this case, the evaluation of $\Delta\text{SL}_{\text{Mass}}$ by the sum of contributions
from continental mass sources (Eq. 5 and 6) needs assumptions on the proportions that end up in the
specific ocean domain (e.g., Tamisiea et al., 2011), that is, on the geographical distribution of water
mass change per surface area, $\Delta\kappa_{\text{source}}$, induced by these continental sources. Based on such assumptions,
$\Delta\text{SL}_{\text{source}}$ may be evaluated as
$$\Delta\text{SL}_{\text{source}} = \frac{1}{\rho_{\text{W}}} \langle \Delta\kappa_{\text{source}} \rangle_{\text{Ocean}}. \quad (8)$$
Here we assume $\langle \Delta\kappa_{\text{source}} \rangle_{\text{Ocean}} = \langle \Delta\kappa_{\text{source}} \rangle_{\text{GlobalOcean}}$, where the suffix "GlobalOcean" refers to
the global ocean, as opposed to a more general ocean domain "Ocean". Our assumption is a
simplification of reality. For example, the gravitationally consistent redistribution of ocean water
induces geographically dependent sea-level fingerprints (Tamisiea et al., 2011).

## 2.2 Time series analysis

The budget assessment is based on anomaly time series $z(t)$ of state parameters, such as sea-level, glacier
mass, etc., where $z(t)$ is the *difference* between the state at epoch $t$ and a reference state $Z_0$. In SLBC_cci,
the reference state $Z_0$ is defined as the mean state over the ten years from Jan 2006 to 2015. This choice
(as opposed to alternative choices such as the state at the start time of the time series) affects plots of
$z(t)$ by a simple shift along the ordinate axis. However, uncertainties of $z(t)$ depend more substantially
on the choice of $Z_0$, which is why they cannot be characterised and analysed without an explicit
definition of the reference state. The epoch $t$ usually denotes a time interval such as a calendar month,
so that $z(t)$ is a mean value over this period.
An alternative way of representing temporal changes is by the rates of change $\frac{\Delta z}{\Delta t}(t)$, where $t$ refers to a
time interval with length $\Delta t$ (e.g. a month or a year) and $\Delta z$ is the change of $z$ during that interval.
Cumulation of $\frac{\Delta z}{\Delta t}(t)$ over discrete time steps gives $z(t)$:
$$z(t) = \sum_{\tau=t_0}^{t} \frac{\Delta z}{\Delta t}(\tau)\,\Delta t. \quad (9)$$
We chose to primarily use the representation $z(t)$ rather than $\frac{\Delta z}{\Delta t}(t)$, that is, we use the evolution of state
rather than its rate of change. The choice is motivated by the characteristics of data products from





satellite altimetry, satellite gravimetry, and Argo floats. They mostly use the representation $z(t)$. Their
differentiation with respect to time amplifies the noise inherent to the observation data.
We analyse the budgets on different temporal scales: First, we analyse the linear trends that arise from
a least squares regression according to
$$z(t) = a_1 + a_2 t + a_3 \cos(\omega_1 t) + a_4 \sin(\omega_1 t) + a_5 \cos(2\omega_1 t) + a_6 \sin(2\omega_1 t) + \varepsilon(t), \qquad (10)$$
where $a_1$ is the constant part, $a_2$ is referred to as the linear trend, or simply the trend, and $\omega_1 = 2\pi$ yr$^{-1}$.
The parameters $a_3$, ..., $a_6$ are co-estimated when considering time series that temporally resolve a
seasonal signal that has not been removed beforehand. We use the trend $a_2$ as a descriptive statistic to
quantify the mean rate of change in a way that is well-defined and robust against noise. The trend $a_2$
thus obtained for different budget elements is then evaluated in budget assessments according to Eq. (1),
(5) and (7).
We apply an unweighted regression in Eq. (10). While a weighted regression may better account for
uncertainties, it would imply that episodes of true interannual variation get different weights in the time
series of different budget elements, so that the trends $a_2$ would be less comparable across budget
elements. As an exception, we apply a weighted regression in one case (the SLBC_cci steric product,
Sect. 3.2.2) where otherwise biases in the early years of the time series would bias the trend.
Second, we analyse the budget on a time series level, that is, we evaluate the budget equations (1), (5)
and (7) for $z(t)$ per epoch. For this purpose, the time series need to be interpolated to an identical monthly
temporal sampling, while for the regression analysis they are left at their specific temporal sampling.

## 2.3 Uncertainty characterisation

Following the 'Guide to the expression of uncertainty in measurement' (JCGM, 2008) we quantify
uncertainties of a measurement (including its corrections) in terms of the second moments of a
probability distribution that "characterises the dispersion of the values that could reasonably be
attributed to the measurand". Specifically, we use the standard uncertainty (i.e. standard deviation, 1-
sigma) to characterise the uncertainty of a measured value. See Merchant et al. (2017) for a recent review
on uncertainty information in the CCI context.
Uncertainty propagation is applied when manipulating and combining different measured values.
Correlation of errors, where present, significantly affects the uncertainty in combined quantities and
careful treatment is required in the context of a budget study in which many millions of measured values
are combined. In this study we have utilised and significantly advanced the characterisation of temporal
error correlations and their accounting in uncertainty propagations, such as for the uncertainty of linear
trends. Where no error correlations are present, the uncertainty of a sum (or difference) of values is the
root sum square of the uncertainties of the individual values. Uncorrelated uncertainty propagation is
applied, in particular, for assessing uncertainties of the sum (or difference) of budget elements since the
data sources for these contributions are mostly independent.
Within this framework for uncertainty characterisation, the uncertainty assessment of each budget
element used a methodology appropriate to the data. Their description in Sect. 3 documents the variety





of approaches, including different ways how error correlations are accounted for explicitly or implicitly.
The requirement to refer $z(t)$ consistently to the mean over the 2006–2015 reference period entailed
adaptations of the uncertainty characterisation for some of the elements.
For each budget element, uncertainties of the linear trends were assessed by the project partners who
contribute the datasets on the budget element. By accounting for temporal error correlations, the trend
uncertainties are typically larger than the formal uncertainty that would arise from the least squares
regression (Eq. 10). Our concept of treating the trend purely as a mathematical functional of the full
time series through which uncertainties can be propagated implies that our evaluated uncertainties in
trends arise only from uncertainties in $z(t)$ and not from statistical fitting effects, such as any true
nonlinear evolution of $z(t)$ or sampling any assumed underlying trend from a short series of data.

## 3 Data sets

### 3.1 Global mean sea-level

*Methods and product*

We use time series of Global Mean Sea-Level (GMSL) anomalies derived from satellite altimetry
observations. For the period Jan 1993 – Dec 2015, the GMSL record is the version 2.0 of the ESA
(European Space Agency) CCI (Climate Change Initiative) 'Sea-level' project (http://www.esa-
sealevel-cci.org/). The CCI sea-level record combines data from the TOPEX/Poseidon, Jason-1/2,
GFO, ERS-1/2, Envisat, CryoSat-2 and SARAL/Altika missions and is based on a new processing
system (Ablain et al., 2015, 2017a; Quartly et al., 2017; Legeais et al., 2018). It is available as a global
gridded 1° × 1° resolution dataset over the 82°N–82°S latitude range. It has been validated using
different approaches including a comparison with tide gauge records as well as to ocean re-analyses and
climate model outputs. The GMSL record is extended with the Copernicus Marine Environment and
Monitoring Service dataset (CMEMS, https://marine.copernicus.eu/) from Jan 2016 to Dec 2016.
The TOPEX-A instrumental drift due to aging of the TOPEX-A altimeter placed in the TOPEX/Poseidon
mission from Jan 1993 to early 1999 was corrected for in the GMSL time series following the approach
of Ablain et al. (2017b). It was derived by comparing TOPEX-A sea-level data with tide gauge data. The
TOPEX-A drift value based on this approach amounts to $1.0 \pm 1.0$ mm yr$^{-1}$ over Jan 1993 to Jul 1995
and to $3 \pm 1.0$ mm yr$^{-1}$ over Aug 1995 to Feb 1999 (see also WCRP, 2018).
For the SLBC_cci project, the gridded sea-level anomalies were averaged over the 65°N–65°S latitude
range. The GMSL time series was corrected for GIA applying a value of -0.3 mm yr$^{-1}$ (Peltier, 2004).
Annual and semi-annual signals were removed through a least square fit of 12-months and 6-months
period sinusoids.
Figure 1a shows the record of GMSL anomalies. The well-known, sustained GMSL rise has a linear
trend of $3.05 \pm 0.24$ mm yr$^{-1}$ over P1. An overall increase of the rate of sea-level rise over the 24 years
is visible (cf. Nerem et al., 2018). The overall GMSL rise is superimposed by interannual variations like
the temporary GMSL drop between 2010 and 2011 by about 6 mm (cf. Boening et al., 2012) with a
subsequent return to the rising path.



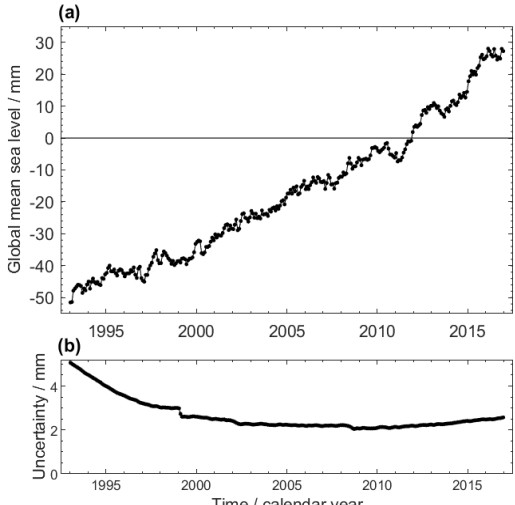

**Figure 1:** (a) Global (65°N to 65°S) mean sea-level time series at its monthly resolution. Changes are expressed with respect to the mean of the reference interval 2006-2015. (b) The assessed standard uncertainties.

*Uncertainty assessment*

Over the recent years, several articles (Ablain et al., 2015, 2017b; Dieng et al, 2017; Quartly et al., 2017; Legeais et al., 2018) have discussed sources of errors in GMSL trend estimation. Ablain et al. (2019) extended these previous studies by considering new altimeter missions (Jason-2, Jason-3) and recent findings on altimetry error estimates. We use the uncertainty assessment by Ablain et al. (2019), which can be summarised as follows:

Three major types of errors are considered in the GMSL uncertainty: (a) biases between successive altimetry missions characterised by bias uncertainties at any given time; (b) drifts in GMSL due to on-board instrumental drifts or long-term drifts such as the error in the GIA correction, orbit etc. characterised by a linear trend uncertainty, and (c) other measurement errors such as geophysical correction errors (wet tropospheric, orbit, etc.) which exhibit temporal correlations and are characterised by their standard deviation. These error sources are assumed to be independent from each other.

For each error source, the variance-covariance matrix over all months is calculated from a large number of random trials (>1000) of simulated errors with a standard normal distribution. The total error variance-covariance matrix is the sum of the individual variance-covariance matrices of each error source. The GMSL uncertainties per epoch are estimated from the square root of the diagonal terms of the total matrix. The covariances are rigorously propagated to assess the uncertainties of multi-year linear trends. In the present study we use standard uncertainties, while Ablain et al. (2019) quote 1.65-sigma uncertainties. Ablain et al. (2019) refer to GMSL anomalies with respect to the mean over a 1993–2017 reference period, while our study uses the 2006–2015 reference period. We neglect the effect of this difference on the uncertainties.



Fig. 1b shows the GMSL anomaly uncertainties per epoch. They are larger during the TOPEX/Poseidon
period (3 mm to 6 mm) than during the Jason period (close to 2.5 mm). This is mainly due to
uncertainties of the TOPEX-A drift correction. Long-term drift errors common to all missions also
increase the uncertainties towards the interval boundaries.

## 3.2 Steric sea-level

### 3.2.1 Ensemble mean steric product for 1993–2016

Since the Argo-based steric product developed within SLBC_cci (see Section 3.2.2 below) does not
cover the full P1 period, for P1 we resort to the existing ensemble mean steric product by Dieng et al.
(2017), updated to include the year 2016. It comprises the following three datsets for the period 1993–
2004: the updated versions of Ishii and Kimoto (2009), NOAA dataset (Levitus et al., 2012) and EN4
dataset (Good et al., 2013). Over the recent years, these datasets integrate Argo data from IPRC
(International          Pacific          Research          Center,
http://apdrc.soest.hawaii.edu/projects/Argo/data/gridded/On_standard_levels/),     JAMSTEC   (Japan
Agency        for        Marine-Earth        Science        and        Technology,
ftp://ftp2.jamstec.go.jp/pub/argo/MOAA_GPV/Glb_PRS/OI/) and SCRIPPS (SCRIPPS Institution of
Oceanography (http://sioargo.ucsd.edu/RG_Climatology.html). Annual and semi-annual signals were
removed. The uncertainty was characterised from the spread between the ensemble members and, where
available, from uncertainties given for the individual ensemble members.
Figure 2 shows the ensemble mean steric time series. It exhibits an overall rise, modulated by
interannual fluctuations which are within uncertainties prior to 2005 but exceed assessed uncertainties
later, e.g. in 2010/2011.

### 3.2.2 SLBC_cci Steric product

Within SLBC_cci, the calculation scheme for the steric sea-level change based on Argo data was
updated from that described by von Schuckmann and Le Traon (2011). Formal propagation of
uncertainty was included following JCGM (2008, their Eq. 13) in which an overall uncertainty estimate
is obtained by propagating and combining the evaluations of uncertainty associated with each source.
*Methods and product*
The steric thickness anomaly for a layer, $l$, of water with density $\rho_l$ is $h'_l = h_l - h_{l,c}$, where $h_{l,c}$ is the
steric thickness of a layer with climatological temperature and salinity and $h_l = \left(\frac{1}{\rho_l} - \frac{1}{\rho_0}\right)\rho_0\,\Delta z_{l0}$ is the
"steric thickness" of the layer *relative to a layer of reference density* $\rho_0$ and reference height $\Delta z_0$. $h'_l$ can
therefore be written in terms of layer density $\rho_l$ and climatological density for the layer $\rho_{l,c}$ as

$$h'_l = \left(\frac{1}{\rho_l} - \frac{1}{\rho_{l,c}}\right)\rho_0\,\Delta z_{l0}\,.$$                  (11)

The monthly mean steric thickness anomaly for layer, $l$, is found as the optimum combination of the
steric thickness anomaly calculations from all the valid profiles in the grid cell for the month. Let the



Earth System
Science
Data

individual anomaly calculations be collected in a vector $x_l$. The optimum estimate is then given by the
following collection of equations:
$$h'_l = w^T_l x_l \qquad (12)$$

$$w_l = \frac{1}{i^T S^{-1}_{x_l} i} S^{-1}_{x_l} i, \qquad (13)$$

where $i$ is a column vector of 1s, $w_l$ is the vector of weights appropriate to a minimum error variance
average, and $S_{x_l}$ is the error covariance matrix of the steric thickness anomaly estimates.
The error covariance matrix, $S_{x_l}$, is needed for the optimal calculation of the monthly average in Eq.
(12), as well as for the evaluation of uncertainty discussed below. To estimate this matrix, we need to
be clear about what "error" means here: it is the difference between the steric thickness anomaly for the
layer from a single profile (Argo, or climatological) and the (unknown) true cell-month mean. This
difference therefore has two components: the measurement error in the profile, characterised by an error
covariance $S_{x_m}$; and a representativeness error arising from variability within the cell-month $S_{x_r}$. The
measurement error covariance is the smaller term and was modelled to be independent between profiles
within the cell (neglecting the fact that on occasion a single Argo float will contribute more than one
profile within a given cell in a month). The representativeness error covariance was modelled assuming
that this error has an exponential correlation form with a length scale of 2.5° and time scale of 10 days.

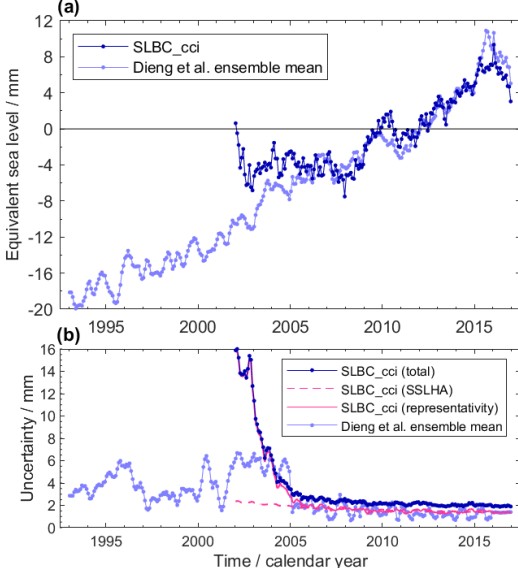


**Figure 2:** (a) Global (65°N to 65°S) mean steric sea-level height anomaly time series at monthly resolution.
Dark blue: dataset generated within SLBC_cci based on Argo data and the CCI SST product. Light blue:
Update of the ensemble mean product by Dieng et al. (2017). Changes are expressed with respect to the
mean of the reference interval 2006-2015. (b) uncertainties assessed for the estimates in (a). Pink curves
(dashed and full lined) show the uncertainty contribution from the SSLHA uncertainty and from the global
representativity uncertainty, respectively.



It is relatively common for there to be layers with no observations, sometimes in the upper ocean and
often at depth. Conditional climatological profiles were used as an additional "observation" to fill in
information for missing-data layers. The climatology of profiles was conditioned by the observed SST
from Merchant et al. (2019), which essentially has negligible sampling uncertainty at monthly cell-
average scales. The SST information constrains the upper-ocean profile to a degree determined by the
vertical correlation of variability, which is variable in time and place according to the mixed layer depth.
The uncertainty in the conditional climatological profile is the variability. Examples of an unconditional
and conditional climatological profile are shown in Fig. 3. For this particular month (August), year
(2003) and location (30.5°N, -9.5°E), the SST is about 2°C below the climatological value. The
conditioning is strong for the upper ~50 m of the ocean, and within this modest depth range the
conditioned profile is realistic given the SST (approximately isothermal over a mixed layer). The
uncertainty is reduced at the surface, where the cell-month SST is well known from the satellite data.
Below about 150 m, the effect of conditioning decays towards zero (conditioned and unconditioned
profiles converge).

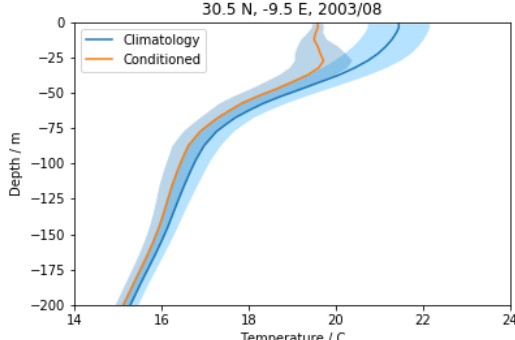


**Figure 3:** Example of effect of conditioning climatology using SST from SST_cci, for a single time and
location (30.5°N, -9.5°E, August 2003). Unconditioned (blue) and conditioned (orange) temperature
profiles with their uncertainty ranges (blue-shaded).
Including SST information slightly reduces uncertainty and affects steric height in the mixed layer often
enough to influence the global mean. Over the period 2005–2018 the trend in steric height is larger using
SST-conditioned climatological profiles than when using a static climatology as the prior. The use of
static climatology to fill gaps in Argo profiles has been shown to cause systematic underestimates of
trends in the literature (e.g., Ishii and Kimoto, 2009). Inclusion of SST-conditioning to the climatology
mitigates the over-stabilising effect.
Steric sea level height anomalies (SSLHA) were calculated for every month from Jan 2002 to Dec 2017
in a global grid of $5° \times 5°$ spatial resolution. For a given cell-month the SSLHA is the sum of the layer-
by-layer estimates of steric thickness anomaly, i.e., $h' = \sum_l h'_l$. By concatenating the vectors $\boldsymbol{w}_l^{\mathrm{T}}$ and $\boldsymbol{x}_l$
for all the layers into a vector of weights $\boldsymbol{w}$ and a vector of thicknesses $\boldsymbol{x}$ we can write the equivalent
equation
$$\boldsymbol{h}' = \boldsymbol{w}^{\mathrm{T}}\boldsymbol{x}. \tag{14}$$



This form makes it more clear how to estimate the uncertainty in the SSLHA, which is
$$u_{h'} = \left(\mathbf{w}^{\mathrm{T}} \mathbf{S}_x \mathbf{w}\right)^{0.5}.$$
(15)

To evaluate the uncertainty we need to formulate $\mathbf{S}_x$. The diagonal blocks corresponding to each layer
in $\mathbf{S}_x$ are the matrices $\mathbf{S}_{x_l}$ that have already been calculated on a layer-by-layer basis. Assumptions about
the error correlations between layers are then required in order to complete the off-block-diagonal
elements of $\mathbf{S}_x$. Conservative assumptions were made:
•   Measurement errors are perfectly correlated vertically in a given profile; this is equivalent to

saying that the sensor calibration bias dominates all other sources of measurement uncertainty

in each profile.

•   Representativity errors are perfectly correlated vertically.
Having obtained cell-month mean SSLHA estimates and associated uncertainty, the global mean steric
sea-level height anomaly, $\Delta \mathrm{SL}_{\mathrm{Steric}}$, is the area-weighted average of the available gridded SSLHA results.
$\Delta \mathrm{SL}_{\mathrm{Steric}}$ was calculated over the range 65°S to 65°N, consistent with other budget elements.
Figure 2a shows the $\Delta \mathrm{SL}_{\mathrm{Steric}}$ time series from the SLBC_cci product. While from 2005 onwards, the
trends of the SLBC_cci product and the ensemble mean product (cf. Table A1) as well as their
interannual behaviour are similar, the SLBC_cci product shows little change prior to 2005. This
difference and its reflection by the uncertainty characterization are discussed further below.
*Uncertainty assessment*
The uncertainties of the available cell-month mean SSLHA estimates were propagated to the $\Delta \mathrm{SL}_{\mathrm{Steric}}$.
In any given month, there are missing SSLHA cells, through lack of sufficient Argo profiles. Using
$\Delta \mathrm{SL}_{\mathrm{Steric}}$ estimated from the available SSLHA cells as an estimate for the global steric sea-level anomaly
introduces a global representativity uncertainty. Moreover, the global sampling errors are correlated
from month-to-month because the sampling distribution evolves over the course of several years towards
near-global representation. To evaluate the global representativity uncertainty and serial correlation, the
sampling pattern of sparse years was imposed on near-complete fields: the standard deviation of the
difference in global mean with the two sampling patterns is a measure of uncertainty. The correlation
between the sample-driven difference in consecutive months was found to be 0.85. The time series of
the global representativity uncertainty, the uncertainty propagated from the gridded SSLHA uncertainty
(which has no serial correlation) and this correlation coefficient combine to define a full error covariance
estimate to be obtained for the $\Delta \mathrm{SL}_{\mathrm{Steric}}$ time series.
Figure 2b shows the two components of uncertainty (global representativity and propagated SSLHA
uncertainty) together with the total uncertainty. The global representativity uncertainty dominates prior
to 2005 and is very large in 2002 and 2003. This reflects how sparse and unrepresentative the sampling
by the Argo network was at that early stage.
Given the large representativity uncertainty prior to 2005, the absence of an increase in the SLBC_cci
steric record during that time is thus understood to arise from global sampling error and is consistent
with the global sampling uncertainty. The SLBC_cci steric time series and the ensemble mean steric
time series are consistent given the evaluated uncertainties throughout the record. In addition, the





evaluated uncertainties for the two time series with their different ways of uncertainty assessment are
remarkably similar for the period of the established Argo network starting from 2005, giving confidence
in the validity of two very distinct approaches to uncertainty characterisation.
The use of a formal uncertainty framework allows separation of distinct uncertainty issues, namely, our
ability to parameterise and estimate the various uncertainty terms, our ability to estimate the error
covariance, and the model for propagation of error at each successive step.
Two aspects of the uncertainty model are recognised to be potentially optimistic: the modelling of
measurement errors as independent between profiles rather than platforms; and the use of only 10 years
for assessing inter-annual variability. Two assumptions are potentially conservative: measurement errors
in salinity and temperature were combined in their worst-case combination; representativity errors in
profiles are assumed to be fully correlated vertically, whereas in reality they are likely to decorrelate
over large vertical separations.
A significant output of the uncertainty modelling of the steric component is the error covariance matrix
for the time series. It enables proper quantification of the change during the time series. We employed
the time-variable uncertainties to determine the linear trend in a weighted regression according to Eq.
(10). Without weighting, the global sampling error prior to around 2005, noted above, would bias any
fitted trend result. Use of the error covariance matrix enables proper quantification of the uncertainty in
the trend calculation by propagating the error covariance matrix through the trend function. Without
this, the serial correlation in the global sampling error would be neglected, and the calculated trend
uncertainty would be an underestimate.

### 453  3.2.3 Deep ocean steric contribution

For the deep ocean below 2000 m depth, the steric contribution was assessed as a linear trend of
$0.1 \pm 0.1$ mm yr$^{-1}$ based on Purkey and Johnson (2010). This deep ocean contribution is included in the
ensemble mean steric product described in Sect. 3.2.2. This deep-ocean component is added to the Argo-
based SLBC_cci steric product described in Sect. 3.2.1 (which is for depths <2000 m) in order to address
the full ocean steric contribution.

## 459  3.3 Ocean-mass change

*Methods and product*
Time series of ocean-mass change (OMC), in terms of anomalies with respect to the 2006-2015 reference
period, were generated from monthly gravity field solutions of the GRACE mission (Tapley et al., 2019).
Similar to previous analyses (Johnson and Chambers 2013, Uebbing et al., 2019) we used spherical
harmonic (SH) GRACE solutions in order to have full control on the methodology and uncertainty
assessment. Greater detail is provided by Horwath et al. (2019).
The following GRACE monthly gravity field solutions series were considered:



•    ITSG-Grace2018 (Mayer-Gürr et al., 2018a, 2018b) from Institut für Geodäsie, Technische
Universität Graz, Austria, with maximum SH degree 60 (data source
ftp://ftp.tugraz.at/outgoing/ITSG/GRACE/ITSG-Grace2018/monthly/monthly_n60)

•    CSR_RL06 from Center for Space Research at University of Texas, Austin, TX, USA,
GFZ_RL06 from Helmholtz Centre Potsdam GFZ German Research Centre for Geosciences,
Germany, JPL_RL06 from Jet Propulsion Laboratory, Pasadena, California, USA, all with
maximum SH degree 60 (data source: https://podaac.jpl.nasa.gov/GRACE).

We chose ITSG-Grace2018 as the preferred input SH solution because it showed the lowest noise level
among all releases considered, with no indication for differences in the contained signal (Groh et al.,
2019b).
Gravity field changes were converted to equivalent water height (EWH) surface mass changes according
to Wahr et al. (1998). The total mass anomaly over an area like the global ocean was derived by spatial
integration of the EWH changes. We used the unfiltered GRACE solutions in order to avoid damping
effects from filtering. A 300 km wide buffer zone along the ocean margins was excluded from the spatial
integration. Around islands, the buffer was applied if their surface area exceeds a threshold, which was
set to 20,000 km² in general and 2,000 km² for near-polar latitudes beyond 50°N or 50°S. The integral
was subsequently scaled by the ratio between total area of the ocean domain and the buffered integration
area. Effects due to the uneven global ocean-mass redistribution are part of the leakage uncertainty
assessment.
Modelled short-term atmospheric and oceanic mass variations are accounted for within the gravity field
estimation procedure (Flechtner et al., 2014; Dobslaw et al., 2013) and are not included in the monthly
solutions. To retain the full mass variation effect, the monthly averages of the modelled atmospheric and
oceanic dealiasing fields were added back to the monthly solutions by using the so-called GAD products
(Flechtner et al., 2014). We subsequently removed the spatial mean of atmospheric surface pressure over
the full ocean domain. Our investigations confirmed findings by Uebbing et al. (2019) on the
methodological sensitivity of this procedure. If the GAD averages were calculated over only the buffered
area, OMC trends would be about 0.3 mm yr$^{-1}$ higher than for our preferred approach.
In order to include the degree-one components of global mass redistribution (not determined by
GRACE) we implemented the approach by Swenson et al. (2008) further developed by Bergmann-Wolf
et al. (2014) which combines the GRACE solutions for degree n≥2 with assumptions on the ocean-mass
redistribution. We also replaced GRACE-based $C_{20}$ components by results from satellite laser ranging
(Cheng et al., 2013, https://podaac-tools.jpl.nasa.gov/drive/files/allData/grace/docs/TN-11_C20_SLR.txt).
GIA implies redistributions of solid Earth masses and (to a small extent) of ocean masses. We corrected
the gravity field effect of GIA-related mass redistributions by using three different GIA modelling
results: The model by A et al. (2013), based on ICE-5Gv2 glaciation history from Peltier (2004); the
model ICE-6G_C (VM5A) by Peltier et al. (2015); and the mean solution by Caron et al. (2018). The
correction was applied on the level of the SH representation. Our preferred GIA correction is the one by
Caron et al. (2018). It is based on the ICE-6G deglaciation history (Peltier et al., 2015), while the model
by A et al. (2013) is based on its predecessor model ICE-5G. Furthermore, while the models by A et al.
(2013) and Peltier et al. (2015) are single GIA models, the solution by Caron et al. (2018) arises as a



weighted mean from a large ensemble of models, where the glaciation history and the solid Earth
rheology have been varied and tested against independent geodetic data to provide probabilistic
information. Table 1 demonstrates the sensitivity of GRACE OMC solutions to the GIA correction.
**Table 1:** OMC linear trends [mm equivalent global mean sea-level per year] over Jan 2003 – Aug 2016
from different GRACE solutions. Each column uses a different GIA correction as indicated in the header
line. The first four lines of data show results from different SH solution series generated within SLBC_cci.
Numbers in brackets are for the ocean domain between 65°N and 65°S. The last two lines show external
products, namely the ensemble mean of the updated time series by Johnson and Chambers (2013) and the
GSFC v2.4 mascon solution. The last column shows the assessed total uncertainty of the trend. The
preferred solution is printed in bold font. The GSFC mascon trend is over the period Jan 2003 – Jul 2016.

|  | GIA from Caron et al. (2018) | GIA from Peltier et al. (2015) | GIA from A et al. (2013) | Uncertainty |
|---|---|---|---|---|
| **ITSG-Grace2018** | **2.19 (2.18)** | 1.93 (1.87) | 1.99 (1.89) | **0.22 (0.25)** |
| CSR RL06 sh60 | 2.17 (2.16) | 1.91 (1.86) | 1.97 (1.87) | 0.22 (0.25) |
| GFZ RL06 sh60 | 2.10 (2.12) | 1.84 (1.81) | 1.90 (1.83) | 0.22 (0.25) |
| JPL RL06 sh60 | 2.19 (2.19) | 1.93 (1.88) | 1.99 (1.90) | 0.22 (0.25) |
| Chambers ensemble | n/a | n/a | 2.17 | n/a |
| GSFC v2.4 mascons | n/a | n/a | 2.25 (2.12) | n/a |


Figure 4a shows our preferred time series of the mass contribution to sea level (see Fig. 10 for a time
series where the seasonal signal is subtracted). The overall trend at $2.19 \pm 0.22$ mm yr$^{-1}$ over P2 is
superimposed by a seasonal signal with 10.3 mm amplitude of annual sinusoid and by interannual
variations like a drop by about 6 mm sea level equivalent between 2010 and 2011.

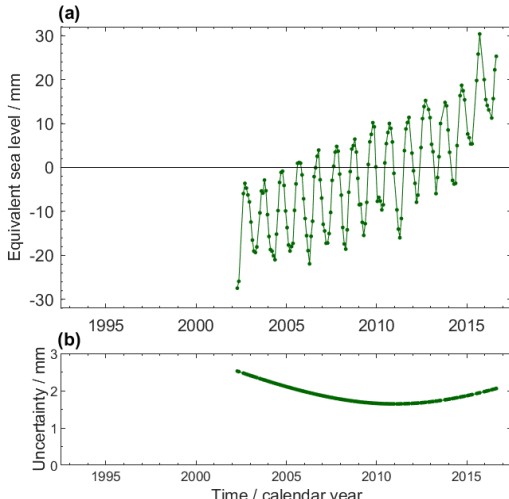


**Figure 4: (a)** Ocean-mass component of GMSL change, derived from the ITSG-Grace2018 spherical
harmonic GRACE monthly solutions with a GIA correction according to Caron et al. (2018). Mass change
of the global ocean is expressed in terms of equivalent GMSL change with respect to the mean of the
reference interval 2006–2015. Time series are shown in their original temporal sampling where some
months are missing. **(b)** Uncertainties assessed for the estimates in (a).





Overall, integrated OMC time series were generated from four series of SH GRACE solutions, using
three GIA corrections (and the option of no GIA correction), for the global ocean and the ocean domain
between 65°N and 65°S. For comparison, we also considered OMC time series from two external
sources: Global OMC time series from CSR, GFZ and JPL SH solutions by Johnson and Chambers
(2013), updated by D. Chambers on 6 November 2017 and made available from
https://dl.dropboxusercontent.com/u/31563267/ocean_mass_orig.txt (accessed 26 Jan 2018); Goddard
Space Flight Center (GSFC) Mascon solutions v02.4 (Luthcke et al., 2013), dedicated for ocean mass
research (data source: https://earth.gsfc.nasa.gov/geo/data/grace-mascons). Time series of total OMC
were derived by the weighted integral over all oceanic points using the ocean-land point-set mask
provided with the GSFC Mascon solutions. We strictly used the area information provided with the
GSFC dataset and rescaled the resulting mass change to the standard ocean surface area of $361 \cdot 10^6$ km$^2$
used within SLBC_cci.
*Uncertainty assessment*
The following sources of uncertainty are relevant (cf. Nagler et al., 2018b):
• GRACE errors: Errors in the GRACE observations as well as in the modelling assumptions
applied during GRACE processing propagate into the GRACE products.
• Errors in $C_{20}$ and degree-one terms: Errors in these components, due to their very large scale
nature and possible systematic effects are particularly important for global OMC applications
(cf. Quinn and Ponte, 2010; Blazquez et al., 2018; Loomis et al., 2019).
• The impact of GIA on GRACE gravity field solutions is a significant source of signal and error
for mass change estimates. Current models show strong discrepancies (Quinn and Ponte, 2010;
Chambers et al., 2010; Tamisiea, 2011; Rietbroek et al., 2016; Blazquez et al., 2018).
• Leakage errors arise from the vanishing sensitivity of GRACE to small spatial scales (high SH
degrees). In SLBC_cci, GRACE data were used up to a degree 60 (~333 km half-wavelength).
As a result, signal from the continents (e.g. ice-mass loss) leaks into the ocean domain.
Differences in methods to avoid (or repair) leakage effects can amount to several tenths of
kg m$^{-2}$ yr$^{-1}$ in regional OMC estimates (e.g., Kusche et al., 2016). Our buffering approach does
not fully avoid leakage. Moreover, the upscaling of the integrated mass changes to the full ocean
area is based on the assumption that the mean EWH change in the buffer is equal to the mean
EWH change in the buffered ocean integration kernel.
We adapted the uncertainty assessment approach used for GRACE-based products of the Antarctic Ice
Sheet CCI project (Nagler et al., 2018b). We modelled errors as the combination of two components
distinguished by their temporal characteristics: temporally uncorrelated noise, with variance $\sigma^2_{\text{noise}}$
assumed equal for each month; and systematic errors of the linear trend, with an associated uncertainty
$\sigma_{\text{trend}}$. This model is a simplification as it does not consider autocorrelated errors other than errors that
evolve linearly with time. The uncertainty $\sigma_{\text{total}}(t)$ per epoch $t$ in a time series of mass anomalies $z(t)$
is approximated as
$$\sigma^2_{\text{total}}(t) = \sigma^2_{\text{noise}} + \sigma^2_{\text{trend}} (t - t_0)^2,$$    (16)



where $t_0$ is the centre of the reference interval to which $z(t)$ refers.
The noise was assessed from the GRACE OMC time series themselves as detailed by Groh et al. (2019a).
The de-trended and de-seasonalised time series were high-pass filtered in the temporal domain. The
variance of the filtered time series was assumed to be dominated by noise. This variance was scaled by
a factor that accounts for the dampening of uncorrelated noise variance imposed by the high-pass
filtering. The assessed noise component comprises uncorrelated errors from all uncertainty sources
except for GIA, which is considered purely linear in time.
The systematic errors of the linear trends are assumed to originate from errors in degree-one
components, $C_{20}$, the GIA correction, and from leakage. The related uncertainties were assessed for each
source individually and summed in quadrature. For example, the GIA uncertainty assessment was based
on the small sample of GIA correction options. The standard deviation over the sample of the three GIA
model options was taken as the standard uncertainty of the GIA correction. The same approach was
applied for the degree-one uncertainty and for the $C_{20}$ uncertainty. To estimate the uncertainty that arises
from leakage, in conjunction with buffering and rescaling, we performed a simulation study based on
synthetic mass change data from the ESA Earth System Model (ESM; Dobslaw et al., 2015). The ESM
data were processed according to the settings of the SLBC_cci OMC analysis and the results (simulated
observations) were compared with the OMC that arises from the full-resolution ESM data (simulated
truth). In order to derive statistics for multi-year trends, we calculated linear trends of the simulated
observations and of the simulated truth and of their misfit for every interval of a length between 9 years
and 12 years contained in the ESA-ESM period. The weighted RMS of misfits over all intervals was
taken as the estimate of the leakage error uncertainty.
Results of the uncertainty assessment for the ITSG-Grace2018-based OMC solutions are summarised in
Table 2. Figure 4b shows the time-dependent uncertainties associated to the ocean mass contribution
time series. They reflect their construction by Eq. (16), where away from 2011.0, the uncertainty of the
linear trend contributes an increasing share.
**Table 2:** Assessed uncertainty components for the OMC solutions based on the ITSG-Grace2018 SH
GRACE solutions.

| Uncertainty component | global ocean domain | ocean domain 65°S – 65°N |
|---|---|---|
| Temporally uncorrelated noise | 1.65 mm | 1.77 mm |
| Trend uncertainty degree-one | 0.14 mm yr$^{-1}$ | 0.14 mm yr$^{-1}$ |
| Trend uncertainty $C_{20}$ | 0.05 mm yr$^{-1}$ | 0.07 mm yr$^{-1}$ |
| Trend uncertainty GIA | 0.14 mm yr$^{-1}$ | 0.17 mm yr$^{-1}$ |
| Trend uncertainty leakage | 0.10 mm yr$^{-1}$ | 0.09 mm yr$^{-1}$ |
| Trend uncertainty combined | 0.22 mm yr$^{-1}$ | 0.25 mm yr$^{-1}$ |




## 3.4 Glacier contribution

*Methods and product*

The glacier mass change estimate was derived by updating the global glacier model (GGM) of Marzeion et al. (2012). Annually reported direct mass balance observations (using the glaciological method) are available for only a few hundred of the roughly 215.000 existing glaciers (Zemp et al., 2019). Global-scale geodetic, altimetric and gravimetric observations are limited to the most recent decades (e.g., Bamber et al., 2018). Only at a regional scale and more disperse, geodetic glacier mass changes are available back into the 1960s (e.g. Maurer et al., 2019; Zhou et al., 2018). The overall objective of the model approach is to use observations of glacier mass change for calibration and validation of the glacier model, which then translates information about atmospheric conditions into glacier mass change, taking into account various feedbacks between glacier mass balance and glacier geometry. This enables a reconstruction of glacier change that is complete in time and space, and that has higher temporal resolution than the observations (here, we use monthly output). In our analysis, we included all glaciers outside of Greenland and Antarctica, and separately reconstructed the glacier change for Greenland peripheral glaciers.

As initial conditions, we used glacier outlines obtained from the Randolph Glacier Inventory (RGI) version 6.0 (updated from Pfeffer et al., 2014). The time stamp of these outlines differs between glaciers, but typically is around the year 2000. To obtain results before this time, the model uses an iterative process to find that glacier geometry in the year of initialisation (e.g., 1901) that results in the observed glacier geometry in the year of the outline's time stamp (e.g., 2000) after the model was run forward.

The model relies on monthly temperature and precipitation anomalies to calculate the specific mass balance of each glacier. It uses the gridded climatology of New et al. (2002) as a baseline. Here, we used seven different sources of atmospheric conditions (as well as their mean) as boundary conditions (Harris et al., 2014; Saha et al., 2010; Compo et al., 2011; Dee et al., 2011; Kobayashi et al., 2015; Poli et al., 2016; Gelaro et al., 2017). Temperature is used to estimate the ablation of glaciers following a temperature-index melt model, and to estimate the solid fraction of total precipitation, which is used to estimate accumulation. Glacier area and length change are estimated following mass change based on volume-area-time scaling, allowing for a delayed response of glacier geometry to glacier mass change. A detailed description of the model is provided by Marzeion et al. (2012).

There are four global model parameters that need to be optimised: (i) the air temperature above which melt of the ice surface is assumed to occur; (ii) the temperature threshold below which precipitation is assumed to be solid; (iii) a vertical precipitation gradient used to capture local precipitation patterns not resolved in the forcing datasets; and (iv) a precipitation multiplication factor to account for effects from (among other processes) wind-blown snow and avalanching, which are not resolved in the forcing dataset. For each of the eight forcing datasets cited above, we performed a multi-objective optimisation for these four parameters, using a leave-one-glacier-out cross validation to measure the model's performance on glaciers for which no mass balance observations exist. We used annual in-situ observations from about 300 glaciers, covering a total of almost 6000 mass balance years (WGMS, 2018). In the optimisation, the temporal correlation of observed and modelled mass balances is

maximised, the temporal variance of modelled mass balances is brought close to that of observed mass
balances (aiming for a realistic sensitivity of the model to climate variability and change), and the model
bias is minimised (to avoid an artificial trend in modelled glacier mass). Using the mean of the seven
atmospheric datasets described above results in the overall best model performance. Compared to the
results in Marzeion et al. (2012), the correlation of annual glacier mass change was increased from 0.60
to 0.64, the bias was changed from 5 kg m$^{-2}$ to -4 kg m$^{-2}$ (both statistically indistinguishable from zero),
and the ratio of the temporal variance of modelled and observed mass balances was improved from 0.83
to 1.00.
The model output for each glacier is aggregated on a regular 0.5° by 0.5° grid, where the mass change
of each glacier is assigned to the grid cell that contains the glacier's centre point, even if the glacier
might cover several grid cells (the GGM does not calculate the spatial distribution of mass changes of a
glacier, so that a more accurate spatial assignment to the grid is not possible). Regional or global values
of glacier mass change were obtained by summing over the region of interest.
Figure 5a shows the global glacier contribution to GMSL anomalies (see Fig. 10 and 12 for time series
after subtraction of the seasonal signal). The glacier contribution has a linear trend of 0.64 ± 0.03
mm yr$^{-1}$ over P1, where the positive rate increases from the first half to the second half of the period.
Interannual variations are less pronounced than for other budget elements.

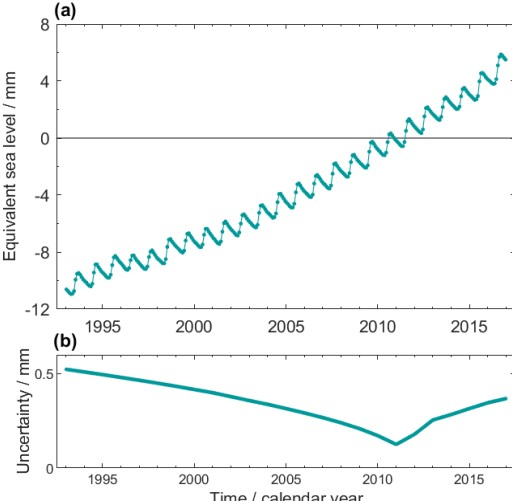

**Figure 5:** (a) Global glacier mass contribution to GMSL assessed by the GGM at a monthly resolution.
Peripheral glaciers in Greenland and Antarctica are not included. Glacier mass change is expressed in terms
of equivalent GMSL change with respect to the mean of the reference interval 2006–2015. (b)
uncertainties assessed for the estimates in (a).
*Uncertainty assessment*
The root mean square error obtained during the cross validation was propagated through the model.
Since the evaluation of the model results does not indicate any temporal or spatial correlation of the





model errors, the uncertainty of temporal and spatial mass change aggregations was calculated assuming
independence of the model errors, i.e. by taking the root of the summed squares of each glacier's (and
year's) uncertainty.
Uncertainties of mass anomalies with respect to the mean over the 2006–2015 interval were
approximated by uncertainties of anomalies with respect to the centre of the interval, 2011.0.
Uncertainties of yearly mass change rates were aggregated (as root sum square) forward or backward
from 2011.0 to the specific epochs. Figure 5c show the uncertainties per epoch, reflecting this
aggregation from 2011.0. Uncertainties of multi-year linear trends were calculated as follows: The
uncertainties of yearly rates of mass change were aggregated in time over the interval of interest, leading
to an uncertainty of cumulated mass change over the interval of interest, which was subsequently divided
by the length of the interval. That is, the trend uncertainty was calculated as the root sum square of
yearly rate uncertainties, divided by the interval length.

## 3.5 Greenland contribution

Changes of land ice masses in Greenland comprising the GrIS and peripheral glaciers are assessed in
two ways: by GRACE (Sect. 3.5.1) and by a combination of satellite altimetry for the GrIS and glacier
modelling for the peripheral glaciers (Sect. 3.5.2, 3.5.3). Results from those complementary assessments
shown in Fig. 6 are collectively discussed at the end of this section.

### 3.5.1 GRACE-based estimates

The GRACE-based product developed at DTU Space within the Greenland Ice Sheet CCI project is used
to provide mass change estimates for the GrIS from GRACE monthly gravity field solutions. The quasi-
monthly GRACE-based mass anomaly estimates (grids and basin time series) are available from
https://climate.esa.int/en/projects/ice-sheets-greenland/. Comprehensive descriptions and references
are given by Barletta et al. (2013), Sørensen et al. (2017), Horwath et al. (2019) and Mottram et al.

681    (2019).

*Methods and product*

An inversion technique was used to obtain monthly mass anomalies for entire Greenland from each of
the available GRACE monthly solutions, with the approach descried in Barletta et al. (2013). An
icosahedral grid of point masses, each representing an area of ~20 km radius, was inverted in order to
fit the gravity observations at the satellite altitude. The limited ~300 km resolution of GRACE monthly
solution requires inversion for ice mass changes over the whole GrIS including peripheral glaciers,
whose contribution cannot be isolated independently. For this work the CSR RL06 monthly solutions
was used, with a maximum degree and order of 96, and prior to the inversion the prescribed $C_{20}$ and
degree-one corrections were applied, together with an anisotropic filtering (DDK3 from Kusche et al.,
2009). Mass changes of glaciers outside Greenland were co-estimated to minimise their leakage into the
Greenland ice mass change estimates.

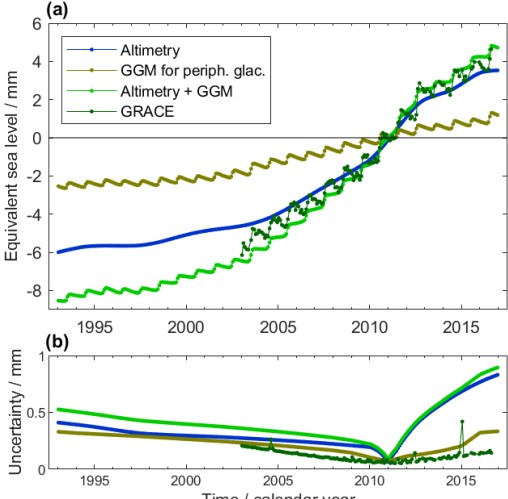


**Figure 6:** (a) Greenland ice mass contribution to GMSL assessed from GRACE (dark green) and from the
combination of altimetry and GGM (light green). The altimetry-based assessment for the ice sheet (blue)
and the GGM-based assessment for the peripheral glaciers (brown) are also shown. Ice mass change is
expressed in terms of equivalent GMSL change with respect to the mean of the reference interval 2006–
2015. GRACE-based time series are shown in their original temporal sampling where some months are
missing. (b) uncertainties assessed for the estimates in (a).

Our inversion did not include a GIA correction. We separately calculated the effect of GIA on our
inversion. Based on the Caron et al. (2018) GIA solution (chosen for consistency with the OMC
estimate, cf, Sect. 3.3) we obtained a GIA effect of 7.5 Gt yr$^{-1}$. This linear trend was subtracted from the
time series.
*Uncertainty assessment*
GRACE based products are provided with error estimates based on the approach developed by Barletta
et al. (2013). The uncertainties were propagated from the errors in GRACE monthly solutions, leakage
errors due to GRACE limited spatial resolutions, and errors in the models used to account for degree-
one contributions and the GIA correction. In detail, the uncertainty related to GRACE solutions was
obtained in a Monte-Carlo-like approach, with 200 simulations for Stokes coefficients selected from a
zero-mean normal distribution, and the standard deviation from the GRACE CSR RL06 Level 2
solution.
3.5.2 Altimetry-based estimates
*Methods and product*
The surface elevation changes estimates are based on satellite radar altimeter observations for the period
1992–2017, and include data from the missions ERS-1, ERS-2, Envisat and Cryosat-2. The temporal
evolution of surface elevation is estimated by a combination of cross-over, repeat-track and least-square





methods covering the entire GrIS and for the entire time span covered by the different missions, on a 5 km common uniform grid (Sørensen et al., 2018). The different characteristics of the missions (the conventional radar altimetry of the ERS-1, ERS-2 and Envisat missions, and the novel SAR Interferometric altimetry of Cryosat-2), and the different orbital characteristics call for special care in the combinations of the different datasets, and in the determination of the uncertainties.

Before an ice sheet wide estimate of volume change can be converted into ice sheet mass balance, contributions which are not related to ice-mass change must be corrected for. These contributions include factors such as changes in firn compaction rates, GIA, and elastic uplift. Such a correction method was first applied for satellite ICESat lidar observation (Sørensen et al., 2011). As Ku-band radar altimetry is subject to weather-induced changes in subsurface penetration depth of snow-covered areas (Nilsson et al., 2015), we here chose to apply a different calibration procedure instead of the direct correction fields. The calibration period is the era of ICESat (2003–2009), where the ICESat laser altimeter provides precise estimates of surface elevation change without surface penetration, and ENVISAT provides similar estimates subject to surface penetration. The spatial differences between the ICESat and ENVISAT mass estimates provides the input for a calibration field (initial radar-volume mass balance, Simonsen et al., 2021, which can be applied to the full time series of elevation changes based on satellite radar altimeter observations for the period 1992–2017. This approach follows that of Simonsen et al., 2021.

Following the calibration procedure described above, we computed monthly grids of mass change rates at $100 \times 100$ km$^2$ resolution for the entire main GrIS. The peripheral glaciers were excluded from the grid. For each epoch, the mass rates of the grid cells were added to derive monthly mass change rates of the entire ice sheet. Time series of ice sheet mass anomalies with respect to the reference interval 2006–2015 were then generated by cumulating the mass change rates in time and subtracting the mean over 2006–2015 from the cumulated time series.

The monthly grids were derived by applying a temporal window to aggregate the radar observations. For the ERS-1, ERS-2, and ENVISAT mission, this window is 5 years long. For CryoSat-2 the window is 3 years long. The monthly grids are referred to the centre of the time window. This result in a smoothening of the time series to resolve climatic changes and not seasonal weather.

*Uncertainty assessment*

The error of the traditional altimetry-based mass-change estimates originates from different sources: uncertainty in the interpolation from point changes to ice sheet wide changes, uncertainty in the bedrock movement and in the firn compaction model, uncertainties due to the neglect of basal melt contributions, and of the possible ice accumulation above the Equilibrium Line Altitude due to ice dynamics. For observations from radar altimetry, an additional source of uncertainty is the changing radar penetration in the firn column. The latter was circumvented by the calibration approach applied here.

The overall uncertainty in the altimetry-derived mass change time series is provided as a conservative estimate based on converting the radar altimetry volume error into mass by ascribing ice densities to all grid cells. This estimate is assumed to be slightly overestimating the combined error of the five error sources.





Uncertainties of cumulated mass changes (in space as well as in time) were derived as follows: For the
cumulation in space, standard uncertainties from all grid cells were added linearly. For the cumulation
in time, uncertainties of mass change rates were aggregated (as root sum square) forward or backward
from 2011.0 to the specific epochs. Uncertainties of multi-year trends were calculated as the aggregated
uncertainties of mass change rates over the interval of interest, divided by the interval length.

### 761     3.5.3 Altimetry-GGM combination

Unlike the GRACE-based assessment for Greenland, the altimetry-based assessment does not include
Greenland peripheral glaciers. We therefore take the sum of the altimetry-based estimates for the ice
sheet and GGM-based estimates for the peripheral glaciers to represent the total ice mass changes in
Greenland. The GGM methods and products and the related uncertainty assessments described in Sect.
3.4 were applied. The uncertainties of the sum of the two products were calculated as the root sum square
of the uncertainties of the two summands.
The synthesis of assessed Greenland GMSL contributions in Fig. 6a shows that both the proper ice sheet
and peripheral glaciers contribute significantly ($0.43 \pm 0.04$ mm yr$^{-1}$ and $0.17 \pm 0.02$ mm yr$^{-1}$,
respectively, over P1). The rates of change vary interannually, peaking in 2011 and 2012. This is
consistently reflected in the GRACE-based estimate and in the altimetry-GGM combination. The
altimetry-GGM combination shows a somewhat larger trend over P2 than the GRACE-based estimate
($0.89 \pm 0.07$ mm yr$^{-1}$ versus $0.78 \pm 0.02$ mm yr$^{-1}$) and does not resolve the annual cycle in the same way
as GRACE, as the annual cycle is not resolved in the altimetry-based time series. The time-variable
uncertainties of the altimetry-based and GGM-based time series (Fig. 6b) reflect the cumulation of
uncertainties of rates of change backward and forward from the reference interval centre. The
uncertainties of the GRACE-based time series reflect the superposition of a linear trend uncertainty and
an individual uncertainty for each monthly GRACE solution.

### 779     3.6 Antarctic contribution

Mass changes of the AIS are assessed in two ways: by GRACE (Sect. 3.6.1) and by satellite radar
altimetry (Sect. 3.6.2). The results from the complementary assessments shown in Fig. 7 are collectively
discussed at the end of this section. The contribution from Antarctic peripheral glaciers is discussed in
Sect. 3.8.

### 784     3.6.1 GRACE-based estimates

The GRACE-based product developed at TU Dresden within the Antarctic Ice Sheet CCI project is used
to provide mass change estimates for the AIS from GRACE monthly gravity field solutions (Horwath
and Groh, 2016; Nagler et al., 2018a, 2018b). Quasi-monthly GRACE-based mass anomaly estimates
(grids and basin time series) are available from https://climate.esa.int/en/projects/ice-sheets-
antarctica/ or https://data1.geo.tu-dresden.de/ais_gmb.

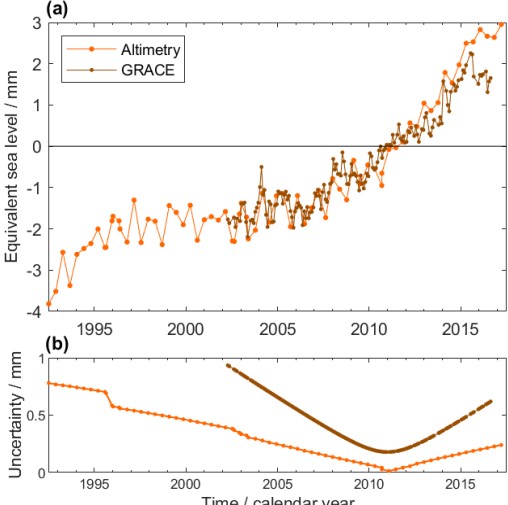

**Figure 7:** (a) Antarctic Ice Sheet mass change contributions to GMSL from GRACE (dark orange) and
altimetry (orange). Ice mass change is expressed in terms of equivalent GMSL change with respect to the
mean of the reference interval 2006–2015. The temporal sampling is quasi-monthly (with a few months
missing) for the gravimetric time series and 140-daily (with a few shorter time increments) for the altimetric
time series. (b) Uncertainties assessed for the estimates in (a).

*Methods and product*

The Antarctic Ice Sheet GRACE-based products were derived from the SH monthly solution series by
ITSG-Grace2016 by TU Graz (Klinger et al., 2016; Mayer-Gürr et al., 2016) following a regional
integration approach with tailored integration kernels that account for both the GRACE error structure
and the information on different signal variance levels on the ice sheet and on the ocean (Horwath and
Groh, 2016). GIA was corrected according to the regional GIA model by Ivins et al. (2013)

*Uncertainty assessment*

The uncertainty assessment (Nagler et al., 2018b) is analogous to that described for the GRACE OMC
assessment in Sect. 3.3. For the AIS, the dominant source of uncertainty is the GIA correction.
Uncertainties in the degree-one components and the $C_{20}$ component of the gravity field are also
important.

### 3.6.2 Altimetry-based estimates

*Methods and Data Product*

We computed Antarctica mass change from 1992 to 2017 using observations from four different satellite
radar altimetry missions – ERS-1, ERS-2, ENVISAT and CryoSat-2 – following the methodology
described by Shepherd et al. (2019). For each mission, we computed elevation change from repeated
elevation measurements during fixed epochs of 140 days on a polar stereographic grid using a plane fit
method (McMillan et al., 2016). We applied a backscatter correction to remove the short-term



fluctuations in elevation change correlated with changes in backscatter and we combined the time series
from different missions together by applying a cross-calibration technique. To convert our elevation
change time series into a mass change time series, we first identified areas of ice dynamical imbalance
in order to discriminate between changes occurring at the density of snow and ice. We defined these
regions as areas with persistent elevation change that is significantly different from firn thickness change
estimates derived from a semi-empirical firn densification model (Ligtenberg et al., 2011). Areas of
accelerated rate of ice thickness change were allowed to evolve through time. Based on this empirical
classification, we converted our elevation change time series to a mass change time series by using a
density of 917 kg m$^{-3}$ in areas classified as ice and using spatially-varying snow densities from the firn
densification model in areas classified as snow. The mass anomalies for the WAIS, the EAIS and the
APIS at a 140-day resolution from 1992 to 2016 are available from http://www.cpom.ucl.ac.uk/csopr/.
*Uncertainty assessment*
We assessed the uncertainties of our elevation change time series and convert them to a mass change
uncertainty using the same time-evolving mask of ice dynamical imbalance areas described in the
previous section. At each epoch, we estimated the overall error of our elevation change as the sum in
quadrature of systematic errors, time-varying errors, errors associated to the calibration between the
different satellite missions and errors associated with snowfall variability. The systematic errors refer to
errors that affect the long-term elevation change trend. These may arise from short-term changes in the
snowpack properties or from short-lived accumulation events that may not be accounted for in our plane
fit model. We quantified the systematic errors as the standard error of the long-term rate of elevation
change. The time-varying error refers to errors in the satellite measurements that might hinder our ability
to measure elevation change at one particular epoch due to the measurement's precision or non-uniform
sampling. We calculated these errors as the average standard error of elevation measurements. The inter-
satellite biases uncertainties were computed as the standard deviations between modelled elevations
during a two-year period centred on each mission overlap. Finally, we quantified the snowfall variability
uncertainty based on estimates from a regional climate model.
Cumulated mass changes and their uncertainties were originally generated with respect to the reference
epoch 1993.0, separately for the East Antarctic Ice Sheet (EAIS), the West Antarctic Ice Sheet (WAIS),
and the Antarctic Peninsula (APIS). To refer the product to the reference interval 2006–2015, we
subtracted the respective mean from the mass anomaly time series. We calculated uncumulated
uncertainties by taking the differences between the uncertainties of consecutive epochs. We re-
cumulated these uncertainties with respect to the centre of the reference interval, 2011.0, by linearly
cumulating the uncumulated uncertainties, forward or backward, from 2011.0. Uncertainties of linear
trends were calculated by linearly cumulating the uncumulated uncertainties over the interval of interest
and division by the interval length. Uncertainties for the mass changes of the entire AIS were calculated
as the root sum square of uncertainties for EAIS, WAIS and APIS.
Figure 7a shows the AIS GMSL contributions from both the GRACE-based and the altimetry-based
assessment. Over P1 (assessed from altimetry), the AIS contribution to GMSL is $0.19 \pm 0.04$ mm yr$^{-1}$.
Rates of change are much smaller from 1995 to 2006 and larger from 2006 onwards. Mass losses are
dominated by mass losses in West Antarctica due to changing ice flow dynamics (cf. Shepherd et al.,
2018). Over P2, the evolution of the AIS GMSL contribution from altimetry and GRACE is similar,



with linear trends at 0.34 ± 0.04 mm yr⁻¹ and 0.27 ± 0.11 mm yr⁻¹, respectively, overlaid by noise as
well as common interannual signal. Time-dependent uncertainties shown in Figure 7b reflect the
cumulation with respect to the reference interval centre in the case of the altimetry-based anomalies, and
the model analogous to Eq. (16) for the GRACE-based estimates.

### 3.7 Land water storage

*Methods and product*
The LWS contribution is assessed with the global hydrological model (GHM) WaterGAP (Döll et al.,
2003; Müller Schmied et al., 2014) in its latest version, WaterGAP2.2d (Müller Schmied et al., 2021).
The model simulates daily water flows and water storage anomalies including the effects of human water
use on a 0.5° × 0.5° grid (55 km by 55 km at equator and ~3000 km² grid cell) covering the whole land
area except for Antarctica (we excluded model outputs over Greenland to avoid double-counting). Note
that the Caspian Sea is not comprised in the model grid (based on the WATCH-CRU land-sea mask)
and thus not included in the assessment of the LWS component. Water flows are routed through a series
of individual water storage compartments (Fig. 2 in Müller Schmied et al., 2021). Following the stream
network defined by the global drainage direction map DDM30 (Döll and Lehner 2002), streamflow is
laterally routed until reaching the ocean or an inland sink. The model is calibrated against observed
mean annual streamflow at 1319 gauging stations (Müller Schmied et al., 2021). LWS anomalies
(LWSA) are the aggregation of the anomalies in all individual water storage compartments:
$$LWSA = SnWSA + CnWSA + SMWSA + GWSA + LaWSA + ReWSA + WeWSA + RiWSA \quad (17)$$
where WSA are water storage anomalies in snow (Sn), canopy (Cn), soil moisture (SM), groundwater
(G), lake (La), reservoir (Re), wetland (We) and river (Ri) storages. The model does not account for
anomalies related to glacier mass variations. Land areas that in reality are covered by glaciers are
represented as non-glacier-covered land areas where hydrological processes (evapotranspiration, runoff
generation, groundwater recharge etc.) are simulated. In terms of OMB assessment, adding the glacier
contribution (Sect. 3.4) and the LWS contribution has the implication of "double-counting" the land
areas covered by glaciers, which are then included in both contributions. In a recent study (Cáceres et
al., 2020), time series of glacier mass variations computed by the GGM of Marzeion et al. (2012) were
integrated as an input to WaterGAP; this resulted in a non-standard version of the model that explicitly
accounts for glaciers. The aggregated water storage anomalies computed by this model version were
compared to the result of adding LWSA computed by the standard WaterGAP and anomalies related to
glacier mass variations computed by the GGM. The comparison of these two approaches showed that
the impact of double-counting glacier-covered areas is insignificant at global scale.
Human water use is accounted for through the representation of the impact of water impoundment in
man-made reservoirs and of net water abstractions (i.e. total abstractions minus return flows) on water
flows and storages. The reservoir operation algorithm implemented in WaterGAP is a slightly modified
version of the generic algorithm of Hanasaki et al. (2006) (Döll et al., 2009). Based on a preliminary
version of the Global Reservoir and Dam (GRanD) data base (Lehner et al., 2011), the model accounts
for the largest 1082 reservoirs. The reservoir filling phase is simulated based on the first operational
year and the storage capacity. Net water abstractions are simulated for five water use sectors (irrigation,





livestock farming, domestic use, manufacturing industries and cooling of thermal power plants) and
subsequently subtracted from the surface water and groundwater storage compartments (Müller
Schmied et al., 2021; Döll et al., 2014).
In the framework of this study, we used monthly globally-averaged (over 64432 0.5° by 0.5° grid cells)
LWSA time series extending from Jan 1992 to Dec 2016. Anomalies are relative to the mean over the
period Jan 2006 to Dec 2015. The model was forced with daily WATCH Forcing Data methodology
applied to ERA-Interim data (WFDEI, Weedon et al., 2014). Two different variants of this climate
forcing were used. In one of them, precipitation was bias corrected using monthly precipitation sums
from the Global Precipitation Climatology Centre (GPCC, Schneider et al., 2015) and, in the other one,
it was bias corrected using monthly precipitation sums from the Climate Research Unit (CRU, Harris et
al., 2014); hereafter, we refer to these climate forcings as WFDEI-GPCC and WFDEI-CRU,
respectively. In addition, we considered two different assumptions in relation to consumptive irrigation
water use in groundwater depletion regions. Typically, consumptive irrigation water use is calculated
by assuming that crops receive enough water for actual evapotranspiration to be equivalent to the
potential evapotranspiration value (Döll et al., 2016). We assumed consumptive irrigation water use to
be either optimal (i.e. 100% of water requirement) or 70% of optimal in groundwater depletion areas
(for more details, see Döll et al., 2014). Consequently, an ensemble of four LWSA time series
corresponding to two climate forcings and two irrigation water use variants was considered. The
unweighted mean of the four ensemble members was used in the SLB assessment.
A comparison of the monthly time series of total water and ice storage anomaly (CMC) over the
continents (except Greenland and Antarctica) as derived from GRACE and from the non-standard
WaterGAP version with glacier integration showed a very good fit, with a modelling efficiency of 0.87
(Cáceres et al., 2020). The GRACE trend during 2003–2016, however, was 26% weaker than the trend
from the non-standard WaterGAP version. More recently this difference was significantly reduced after
the GRACE analysis for continental total water storage was made more consistent with the GRACE
OMC analysis (Gutknecht et al., 2020).
Figure 8a shows the monthly time series of LWSA contribution to GMSL. It is characterised by the
highest seasonal amplitude of all ocean-mass contributions due to seasonal climate variations. (See Fig.
12 for a time series where the seasonal signal is subtracted.) The overall positive trend ($0.40 \pm 0.10$
mm yr$^{-1}$ over P1) is caused mainly by groundwater and surface water depletion that more than balances
increased land water storage due to the filling of new reservoirs.
*Uncertainty assessment*
Uncertainties are characterised by the spread between the four model runs. For each month, the standard
deviation of the values from the four time series was taken as the standard uncertainty. Figure 8b shows
these time-variable uncertainties of the LWSA. They reflect month-to-month differences in the spread
between the ensemble members. Since the LWS anomalies are referred to the 2006–2015 mean value
and the four ensemble members show different trend, the uncertainty is lowest around 2011.0 and tends
to increase towards the beginning (1993) and the end (2016). The standard deviation of the linear trends
calculated for each ensemble member was taken as the standard uncertainty of the linear trend of the
ensemble mean.

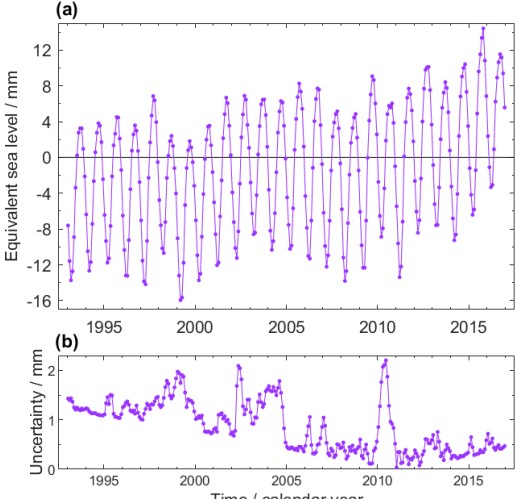


**Figure 8:** (a) Contributions from global land water storage changes (except for Greenland and Antarctica)
to GMSL, assessed by the WaterGAP global hydrology model at its monthly resolution. Water mass change
is expressed in terms of equivalent GMSL change with respect to the mean of the reference interval 2006–
2015. (b) uncertainties assessed for the estimates in (a)

### 3.8 Other contributions and issues

Caspian Sea water storage changes are not included in the WaterGAP model domain and are therefore
not included in our GMSL budget assessment. WCRP (2018) quote this contribution as
$0.075 \pm 0.002$ mm yr$^{-1}$ since 1995 and $0.109 \pm 0.004$ mm yr$^{-1}$ since 2002. Based on GRACE analyses,
Cáceres et al. (2020) estimate the contribution to be $0.066 \pm 0.003$ mm yr$^{-1}$ over 2003–2016, very
similar to the GRACE-based estimate by Loomis and Luthcke (2017), which corresponds to
$0.067 \pm 0.007$ mm yr$^{-1}$ sea-level equivalent over 2003–2014.

Antarctic peripheral glaciers are neither included in the altimetry-based assessment of the Antarctic ice
mass change nor in the GGM assessment. The GRACE-based estimate for Antarctic ice mass changes
was designed to address the ice sheet proper but includes part of the mass changes of peripheral glaciers
in result of the low spatial resolution capability of satellite gravimetry. Gardner et al. (2013) estimate
the Antarctic peripheral glaciers mass loss over 2003–2009 to a value equivalent to
$0.017 \pm 0.028$ mm yr$^{-1}$ GMSL. Zemp et al. (2019) estimate a loss over 2006–2016 at a value equivalent
to $0.04 \pm 0.30$ mm yr$^{-1}$ GMSL.

Changes in atmospheric water content (mainly tropospheric water vapour) are not included in our
assessment. The atmosphere stores around 12,700 Gt of water (Trenberth 2014), or 35 mm sea-level
equivalent. Hartmann et al. (2013) report that the rate of change of tropospheric water vapour content is
very likely consistent with the Clausius–Clapeyron relation (about 7% increase in water content per
Kelvin). This corresponds to an equivalent GMSL effect on the order of -0.03 to -0.05 mm yr$^{-1}$, which
was also obtained by Dieng et al. (2017) from ERA-Interim atmospheric reanalysis results. Interannual





variations of atmospheric water content reported by Dieng et al. (2017) are up to the order of 1 mm
GMSL equivalent.
The elastic deformation of the ocean bottom induced by the present-day global redistribution of water
and ice loads is not accounted for in our GMSL estimate from satellite altimetry (cf. Sect. 2.1). Since
the deformation is downward on average over the global ocean, this omission leads to an
underestimation of relative GMSL rise. Frederikse et al. (2017) estimated the effect for the period 1993–
2014 to be 0.13 mm yr$^{-1}$ for the global ocean and 0.17 mm yr$^{-1}$ for the domain bounded by ±66° latitude,
with higher rates in the second half of the period. Vishwakarma et al. (2020) estimated the effect for
2005–2015 to be 0.11 ± 0.02 mm yr$^{-1}$ for a global altimetry domain buffered along the coasts.
Our conversion from OMC (or ocean-mass contributions) to sea-level change adopts the density of
freshwater. In previous studies, either the density of fresh water or the density of sea water has been
adopted, where both approaches have their justification (cf. Gregory et al., 2019; Vishwakarma et al.,
2020). If we had adopted the sea water density (1028 kg m$^{-3}$), our assessments of mass contributions
would be reduced by 2.7%.

# 4 Ocean-mass budget

We evaluate the OMB according to Eq. (5). We do the assessment for the P2 period (Jan 2003 –Aug
2016, cf. Sect. 2.2). We use the OMC assessment made for the global ocean.

## 4.1 Linear trend

For the elements of the mass budget, we calculated linear trends over P2. We assessed their uncertainties
as explained in Sect. 2.3 and specified for every element in Sect. 3. The results are shown in Table 3.
All components exhibit a significant positive trend, i.e., water mass loss on land. Greenland ice masses
contribute 0.78 ± 0.02 mm yr$^{-1}$ as assessed from GRACE or 0.89 ± 0.07 mm yr$^{-1}$ as assessed from radar
altimetry for the ice sheet and from the GGM for the peripheral glaciers. The glaciers outside Greenland
and Antarctica contribute 0.77 ± 0.03 mm yr$^{-1}$, similar to Greenland. The Antarctic Ice Sheet's
contribution is 0.27 ± 0.10 mm yr$^{-1}$ if assessed from GRACE, and 0.34 ± 0.04 mm yr$^{-1}$ if assessed from
radar altimetry. The trend in land water storage amounts to 0.40 ± 0.10 mm yr$^{-1}$.
The sum of components is 2.19 ± 0.15 mm yr$^{-1}$ and 2.40 ± 0.13 mm yr$^{-1}$, respectively, if the Greenland
and Antarctica contributions are assessed using either GRACE or altimetry. The corresponding trend in
mean global ocean mass according to our preferred GRACE-based solution (ITSG-Grace2018, GIA
correction according to Caron et al., 2018) amounts to 2.19 ± 0.22 mm yr$^{-1}$.
The misclosures of Eq. (5) with combined standard uncertainties are -0.04 ± 0.27 mm yr$^{-1}$ (if using
GRACE for Greenland and Antarctica) and -0.21 ± 0.26 mm yr$^{-1}$ (if using altimetry in Greenland and
Antarctica). Hence, the mass budget in terms of linear trends is closed within the assessed uncertainties.
In view of the systematic uncertainties inherent to several components of the mass budget, we stress that
any closure that is much better than the combined uncertainties does not indicate that the components





are correct at the level of the budget closure, but may just be a coincidence of trend errors compensating
each other.

**Table 3:** Linear trends of the mass budget elements [mm equivalent global mean sea-level per year] for the
interval P2, and their standard uncertainties.

| Budget element | Method | P2: Jan 2003 – | Aug 2016 |
|---|---|---|---|
| Glaciers | GGM | $0.77 \pm 0.03$ | $0.77 \pm 0.03$ |
| Greenland | Altimetry | $(0.68 \pm 0.06)$ | |
| | GGM | $(0.21 \pm 0.03)$ | |
| | Altimetry + GGM | $0.89 \pm 0.07$ | |
| | GRACE | | $0.78 \pm 0.02$ |
| Antarctica | Radar altimetry | $0.34 \pm 0.04$ | |
| | GRACE | | $0.27 \pm 0.11$ |
| Land water storage | WaterGAP | $0.40 \pm 0.10$ | $0.40 \pm 0.10$ |
| Sum of mass contributions | | $2.40 \pm 0.13$ | $2.22 \pm 0.15$ |
| Ocean mass (global) | GRACE | $2.19 \pm 0.22$ | $2.19 \pm 0.22$ |
| Misclosure | | $-0.21 \pm 0.26$ | $-0.04 \pm 0.27$ |

## 4.2 Seasonal component

The inherent monthly resolution of GRACE-based OMC, GRACE-based AIS and GrIS mass changes
and modelled LWS and glacier mass changes allows us to analyse the budget of the seasonal variations
of ocean mass. For this purpose, we analyse the annual cosine and sine amplitudes $a_3$ and $a_4$ of Eq. (3)
just in the way we analysed the linear trend $a_2$ in Sect. 4.1.
Figure 9 shows the results of this analysis. The seasonal amplitudes of GRACE-based OMC and the
sum of assessed contributions are very similar at 10.3 mm and 9.7 mm, respectively. LWS, with an
amplitude of 8.9 mm, is by far the dominant source of seasonal OMC. The phase of GRACE OMC is
approximately 7 days later than the phase of the sum of components. This small offset of phase is within
the uncertainties assessed for the GRACE OMC results, even though the uncertainty assessment was
limited to effects of degree-one, $C_{20}$, and leakage. Errors in the seasonal components of WaterGAP are
another potential source of the phase offset.
The result does not change significantly (by less than 0.15 mm SLE for the annual cosine and sine
amplitudes) if we replace the WaterGAP ensemble mean by one of the individual WaterGAP model
runs or if we replace the ITSG-based GRACE OMC solutions by the CSR-based, GFZ-based, or JPL-
based SH OMC solution generated by the SLBC_cci project. The phase offset between GRACE OMC
and the sum of contributions becomes larger if we replace the SLBC_cci OMC solutions by the
Chambers SH-based OMC solutions or the GSFC mascon solutions (cf. specifications in Sect. 3.3) – see
grey arrows in Fig. 9.



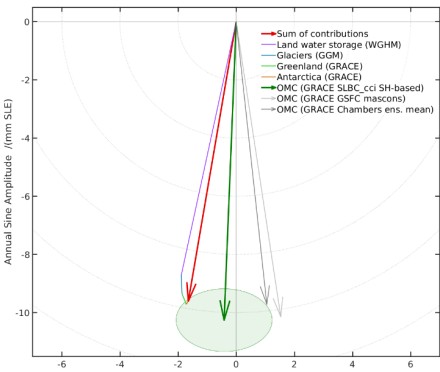


**Figure 9:** Phase diagram of annual sine and cosine amplitudes of elements of the ocean-mass budget. Bold
red vector: sum of contributions in (using GRACE-based estimates for Greenland and Antarctica). Coloured
thin lines: individual contributions (see legend). Bold dark green vector: GRACE ocean-mass change
(OMC) SLBC_cci solution based on GRACE ITSG-Grace2018, together with the uncertainty ellipse. Thin
grey vectors: external GRACE OMC solutions (GSFC mascons, Chambers' ensemble mean, see legend).
The phase difference between the red and the dark green vector corresponds to 7 days.
## 4.3 Monthly time series
Figure 10 illustrates the monthly-sampled time series of the elements of the OMB. The seasonal signal
component, represented by annual and semi-annual harmonic functions was subtracted. The GRACE
OMC (dark green line), and the sum of components (dark red line or magenta line) not only have very
similar trends (cf. Sect. 4.1). They also reflect interannual variations coherently. These interannual
variations overlay the long-term trend and reach amplitudes of 2–3 mm. Clearly, they are dominated by
the LWS contribution. They include a minimum in 2007/2008, a maximum in 2010, with a subsequent
decrease to a minimum in 2011 related to a La-Niña event (Boening et al., 2012). The sequence
continues with an interannual maximum in 2012/2013, a minimum in 2013/2014 and another maximum
in 2015/2016.
Figure 11a shows the OMB misclosure, together with the combined standard uncertainties of all
elements of Eq. (5). The percentages of monthly misclosure values within the 1-sigma, 2-sigma, and 3-
sigma combined uncertainty amount to 65.9%, 94.5% and 100.0% for the time series using the GRACE-
based ice sheet assessment. Similarly, the percentages are 65.9%, 93.3% and 100.0% for the time series
using the altimetry-based ice sheet assessments. These statistics support the realism of the uncertainty
assessment where under the assumption of a Gaussian error distribution one would expect 67.3%,
95.5%, and 99.7% of the values to be within the 1-sigma, 2-sigma, and 3-sigma limits, respectively.

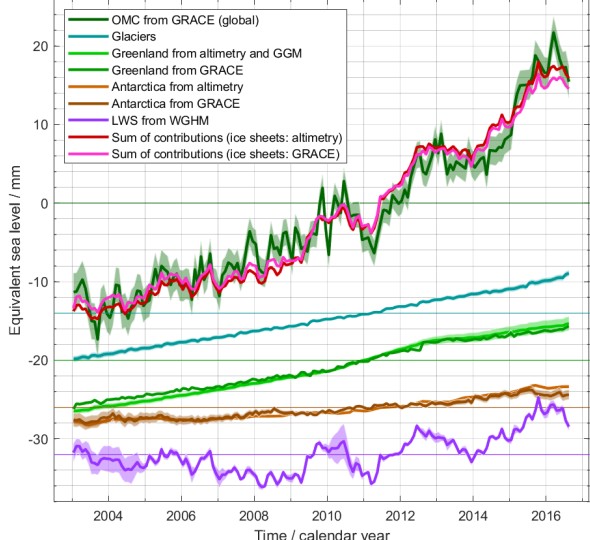

**Figure 10:** Time series of the elements of ocean-mass budget in Jan 2003 – Aug 2016. See legend for attribution of graphs. The GRACE-based time series and the Antarctic altimetry time series were interpolated to monthly sampling. Seasonal variations are subtracted. Each graph shows anomalies with respect to a mean value over 2006–2015. Graphs are shifted arbitrarily along the ordinate axis. Transparent bands show standard uncertainties (except for the red sum-of-contribution graphs).

# 5 Sea-level budget

We consider the two time periods P1 (altimetry era) and P2 (GRACE/Argo era) as introduced in Sect. 2.2. We concentrate on the steric product generated within SLBC_cci (see Sect. 3.2) when analysing the SLB over P2. For P1, which is not fully covered by the SLBC_cci steric product, we resort to the ensemble mean steric product updated from Dieng et al. (2017). The GRACE-based OMC estimates used here are those evaluated for the ocean between 65°N and 65°S. While for the present results of GRACE OMC this makes little difference, it is consistent to the averaging area of GMSL and the steric component.

## 5.1 Linear trend

Linear trends for the elements of the SLB for the two time periods are given in Table 4. The trends were calculated as explained in Sect. 2.2, and uncertainties were assessed as explained in Sect. 2.3 and specified for every element in Sect. 3.

For P1, the observed GMSL trend is $3.05 \pm 0.24$ mm yr$^{-1}$. The sum of individual SLBC_cci v2 components is $2.90 \pm 0.17$ mm yr$^{-1}$. This leaves a misclosure of $0.15 \pm 0.29$ mm yr$^{-1}$.

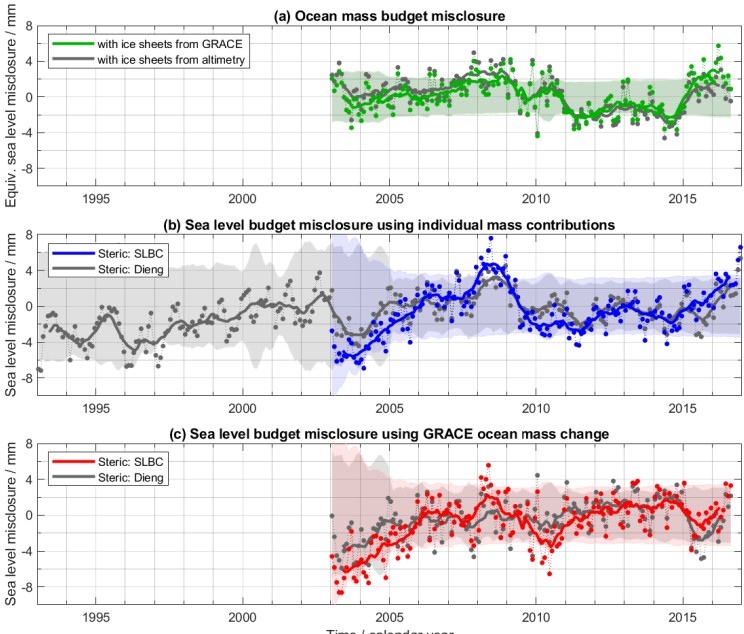

**Figure 11:** (a) Ocean-mass budget misclosure (GRACE-based OMC minus sum of assessed contributions)
for the time series of monthly anomalies of mass budget elements as shown in Fig. 10. Dots: monthly
misclosure for the case of GRACE-based (green) and altimetry-based (grey) ice sheet assessments. Thick
lines: running 12-month mean, for better visibility of interannual features. Shaded bands (green and grey,
almost identical in the figure): combined standard uncertainty (1-sigma) of the monthly misclosure. (b) Sea-
level budget misclosure (GMSL minus sum of contributions) for the time series shown in Fig. 12 using the
individual mass contributions (involving the altimetry-based ice sheet assessments). Dots, lines, shaded
areas have meanings as in subplot a. Blue and grey: results employing the SLBC_cci Steric data product
and the Dieng et al. (2017) ensemble mean dataset, respectively. (c) Same as subplot (b) but with application
of GRACE-based OMC, instead of the sum of assessed mass contributions. Red and grey: results employing
the SLBC_cci Steric data product and the Dieng et al. (2017) data product, respectively.

For P2, the observed GMSL trend is $3.64 \pm 0.26$ mm yr$^{-1}$. The sum of contributions is
$3.37 \pm 0.30$ mm yr$^{-1}$ if OMC is estimated from GRACE. The sum of contributions is
$3.59 \pm 0.22$ mm yr$^{-1}$ and $3.41 \pm 0.23$ mm yr$^{-1}$, if the mass contributions are assessed individually,
involving altimetry-based estimates or GRACE-based estimates, respectively, for the ice sheets. The
three choices of assessing OMC leave misclosures of $0.27 \pm 0.40$ mm yr$^{-1}$, $0.05 \pm 0.34$ mm yr$^{-1}$ and
$0.23 \pm 0.35$ mm yr$^{-1}$, respectively. The trend misclosures are hence positive but below the standard
uncertainty arising from the combined uncertainties of the involved budget elements.

If we used the Dieng et al. (2017) ensemble mean steric product for the P2 SLB assessment, the trend
misclosure remained unchanged, since the steric trend over P2 is equally 1.19 mm yr$^{-1}$ for the two
alternative steric products.





**Table 4:** Linear trends of the sea-level budget elements [mm equivalent global mean sea-level per year] for the intervals P1 and P2, and their standard uncertainties. The estimates of total sea-level, the steric contribution, and the GRACE-based ocean mass contribution refer to the ocean between 65°N and 65°S.

| Budget element | Method | P1: Jan 1993 – Dec 2016 | P2: Jan 2003 – Aug 2016 | | |
|---|---|---|---|---|---|
| Total sea-level | altimetry | 3.05 ± 0.24 | 3.64 ± 0.26 | 3.64 ± 0.26 | 3.64 ± 0.26 |
| Steric component | Dieng | 1.15 ± 0.12 | | | |
| | SLBC_cci + deep steric estimate | | 1.19 ± 0.17 | 1.19 ± 0.17 | 1.19 ± 0.17 |
| Glaciers | GGM | 0.64 ± 0.03 | 0.77 ± 0.03 | 0.77 ± 0.03 | |
| Greenland | (altimetry) | (0.43 ± 0.04) | (0.68 ± 0.06) | | |
| | (GGM) | (0.17 ± 0.02) | (0.21 ± 0.03) | | |
| | Altimetry + GGM | 0.60 ± 0.04 | 0.89 ± 0.07 | | |
| | GRACE | | | 0.78 ± 0.02 | |
| Antarctica | altimetry | 0.19 ± 0.04 | 0.34 ± 0.04 | | |
| | GRACE | | | 0.27 ± 0.11 | |
| Land water storage | WaterGAP | 0.32 ± 0.10 | 0.40 ± 0.10 | 0.40 ± 0.10 | |
| Sum of mass contributions | | 1.75 ± 0.12 | 2.40 ± 0.13 | 2.22 ± 0.15 | |
| Ocean mass (65°N-65°S) | GRACE | | | | 2.18 ± 0.25 |
| Sum of contributions | | 2.90 ± 0.17 | 3.59 ± 0.22 | 3.41 ± 0.23 | 3.37 ± 0.30 |
| Misclosure | | 0.15 ± 0.29 | 0.05 ± 0.34 | 0.23 ± 0.35 | 0.27 ± 0.40 |

The strong limitations of Argo coverage in the years before 2005 are reflected in the uncertainties of the SLBC_cci Steric product (Fig. 3). Since the trend calculation accounts for these uncertainties (cf. Sect. 2.2) the SLBC_cci steric trend is dominated by the data starting from 2005. An alternative accounting for the high pre-2005 uncertainties would be to start the entire SLB assessment from 2005. For an alternative period Jan 2005 – Aug 2016, the trend budget is given in Table A1 (Appendix). For this period, the assessed linear trends of GMSL, the steric component, and the mass component are higher, by about 0.16, 0.07 and 0.20–0.29 mm yr$^{-1}$, than for P2. The conclusions on the budget closure within uncertainties remain unchanged.

## 5.2 Monthly time series

For P1 (altimetry era) our SLB assessment refers to the Dieng et al. (2017) ensemble mean steric product and to the mass component composed from the individual contributions (involving altimetry-based assessments for the ice sheets). Figure 12 shows the monthly time series of the SLB elements. The seasonal signal component is removed. Apart from showing similar linear trends (cf. Sect. 5.1), the observed GMSL (black curve) and the sum of contributions (dark red curve) exhibit largely coherent interannual variations in the second half of P1 starting from 2005. These interannual variations, overlaid on the long-term trend, reach about 3–4 mm amplitudes. As a prominent feature, the La-Niña related local GMSL minimum in 2011 (Boening et al., 2012) arises as a superposition of synchronous variations of a LWS effect and a steric effect.

The associated misclosure time series are shown in Fig. 11b (in grey). Deviations between GMSL and the sum of components are relatively large in the early years 1993–1996. In this period GMSL uncertainties are large (cf. Fig. 1b) due to uncertainties of the TOPEX-A drift correction. In addition, the steric component has large uncertainties in this period and further through 2004, where it is based

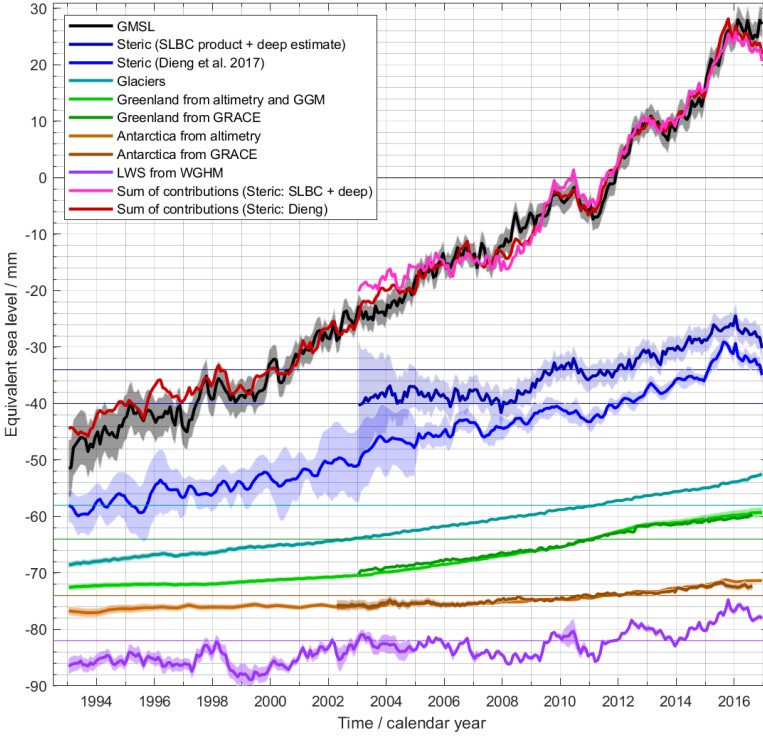

on XBT data and therefore suffers from sparse coverage both geographically and at depth (below 700 m). The monthly misclosure values for P1 are within the 1-sigma and 2-sigma uncertainty band, respectively, for 90.9% and 100.0% of the months. Hence, the misclosure has a narrower distribution than allowed for by the assessed uncertainties.

For P2 (GRACE/Argo era) the SLB analysis may employ the SLBC_cci steric dataset, which is also shown in Fig. 12. Again, interannual variations of GMSL (black curve) and the sum-of-components (magenta curve) agree largely in their sequence of positive and negative deviations from a long-term evolution, with the exception of the early Argo years 2003 and 2004. Figure 11b shows (in blue) the misclosure of the SLB when using this SLBC_cci steric dataset (and the individual mass contribution assessments). The percentage of misclosures within the 1-sigma, 2-sigma and 3-sigma ranges of combined uncertainties are 84.8%, 99.4%, and 100.0%, respectively.

**Figure 12:** Time series of SLB elements involving the individual contributions to ocean-mass change. See legend for attribution of graphs. The sum-of-components graphs use altimetry-based ice sheet assessments. The GRACE-based time series and the Antarctic altimetry time series were interpolated to monthly sampling. Seasonal variations are subtracted. Each graph shows anomalies with respect to a mean value over 2006–2015. Graphs are shifted arbitrarily along the ordinate axis. Standard uncertainties are shown by transparent bands (except for the sum-of-contribution graphs).



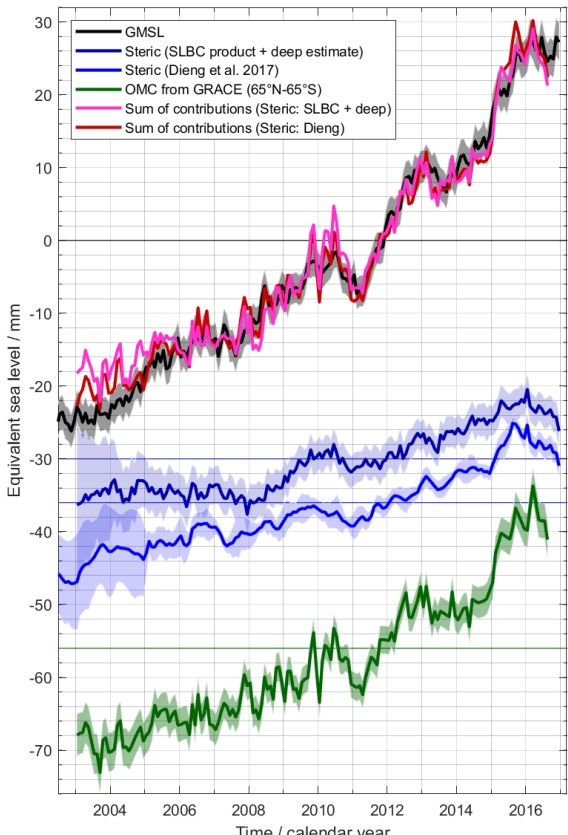


**Figure 13:** Time series of SLB elements involving the GRACE-based assessment of ocean-mass change.
See legend for attribution of graphs. The GRACE-based time series were interpolated to monthly sampling.
Seasonal variations are subtracted. Each graph shows anomalies with respect to a mean value over 2006–
2015. Graphs are shifted arbitrarily along the ordinate axis. Standard uncertainties are shown by transparent
bands (except for the sum-of-contribution graphs).
For the case of using the GRACE-based OMC, the monthly budget assessment over P2 is illustrated in
Fig. 13. While the use of GRACE OMC introduces more month-to-month noise into the sum-of-
components time series than the use of individual mass contributions, the features of interannual
variations discussed above are again coherently reflected in the GMSL and the sum of-contributions.
The related monthly misclosure time series are shown in Fig. 11c. When using the SLBC_cci steric
product, the monthly misclosure values are within the 1-sigma, 2-sigma, and 3-sigma range,
respectively, for 89.6%, 100.0% and 100.0% of the months.



## 6 Attribution of misclosure

We cannot attribute the misclosures in the budgets of linear trends, as the uncertainties on the order of 0.3 mm yr$^{-1}$ in various elements of the OMB and SLB would make such an attribution extremely ambiguous. In contrast, for the interannual features in the misclosure time series of the different budgets (Fig. 11) we can suggest indications on misclosure origins by comparing them among each other and with the interannual variations of the budget elements. Interannual variations are depicted as variations of the running annual means of the misclosure time series, shown as bold curves in Fig. 11.

The OMB misclosure varies interannually between roughly -2 mm and +2 mm (bold curves in Fig. 11a). The SLB misclosure varies interannually between roughly -6 mm and +4 mm (bold curves in Fig. 11b, c) depending on which steric product and which way of estimating OMC are used. Errors of the datasets on GMSL, the steric contribution, GRACE-based OMC, and LWS are most likely responsible for these interannual misclosures. The glaciers and ice sheets time series involve relatively small interannual variations (cf. Fig. 12) so that their errors are unlikely to exceed the sub-millimetre level. The unassessed atmospheric water content contribution (cf. Sect. 3.8) could add to the misclosure, though.

As a starting point, we discuss the SLB misclosure obtained if estimating the steric contribution by the SLBC_cci steric product and estimating the mass component by the sum of mass contributions (Fig. 11b, blue curve). As a first feature, the misclosure moves from -6 mm to 0 mm between mid-2003 and mid-2006, indicating that over this 3-year period the sum of contributions rose 2 mm yr$^{-1}$ less than the altimetry-based GMSL. At least part of this feature is readily explained by the limitations of the SLBC_cci steric product in these early years of the Argo system, as discussed in Sect. 3.2.

As a second prominent feature, the misclosure rises by 4 mm from 2006 to 2008 and falls again by 6 mm from 2008 and 2010. From Fig. 12 we see that this misclosure is related to the fact that the sum of components suggests a temporary slowdown of sea-level rise from 2006 to 2008, while the altimetric GMSL exhibits less of such a slowdown. The SLBC_cci steric time series (Fig. 3, 12) has a feature of fall and rise by 3-4 mm in those 2006–2008 and 2008–2010 periods, and this feature enters the SLB misclosure with negative sign. In addition, the mass budget misclosure (Fig. 11a) has a similar rise and fall by 1 to 2 mm in the same 2006–2008 and 2008–2010 periods. Replacing the individual mass components by GRACE-based OMC reduces the misclosure feature (compare Fig. 11b blue line and Fig. 11c red line). Replacing the SLBC_cci steric product by the Dieng et al. (2017) ensemble mean steric time series further reduces this feature (compare Fig. 11c red line and grey line). This may suggest that between 2006 and 2010 the interannual variations of OMC and the steric component are more accurately represented by GRACE-based OMC and the Dieng ensemble mean, respectively, than by the sum of mass contributions involving modelled LWS and the SLBC_cci steric product.



# 7 Discussion

## 7.1 Budget closure and uncertainties

In all our budget assessments for linear trends, the misclosure is within the 1-sigma range of the combined uncertainties of the budget elements. As a consequence, no statistically significant estimates of missing budget elements can be made based on the present budget assessments. Assessed uncertainties of the trends of various budget elements are on a similarly high level. For example, for P2 the trend uncertainties are 0.26 mm yr$^{-1}$ for GMSL, 0.17 mm yr$^{-1}$ for the steric component, and 0.13 mm yr$^{-1}$ to 0.25 mm yr$^{-1}$ for the mass component, depending on how it is assessed. This prevents us from validating the linear trend of any single component through budget considerations. For example, we cannot easily use budget considerations to decide which GRACE-based OMC trend in Table 1 is most accurate. In cases where the budget of trends is closed much better than the combined uncertainties (e.g., the second data columns in Table 3 and Table 4), this could just result from an incidental compensation of errors in the involved budget elements.

The trends of the individual budget components assessed here for P1 and P2 agree within stated uncertainties with the trends of the same budget components assessed by the IPCC SROCC (Oppenheimer et al., 2019, Table 4.1) for the periods 1993–2015 and 2006–2015, respectively, even if differences between the considered periods may add to the differences between the two assessments. The single exception from this agreement applies to the LWS contribution. While SROCC reports a negative sea-level contribution at -0.21 mm yr$^{-1}$ for 2006–2015 based on GRACE analyses, our WaterGAP results indicate a positive contribution at 0.65 mm yr$^{-1}$ for 2006-2015 (0.40 mm yr$^{-1}$ for P2). However, a new GRACE-based assessment of continental mass change (Cáceres et al., 2020, updated by Gutknecht et al., 2020; cf. Sect. 3.8) corrected for the GGM-based glacier mass trend also determines a positive LWS contribution at 0.31 mm yr$^{-1}$ for 2006–2015 (0.42 mm yr$^{-1}$ for P2), consistent with the WaterGAP results. Assessing the LWS contribution to GMSL remains a challenge.

Our analysis of the OMB and the SLB on a time series basis exploits the intrinsic monthly resolution of almost all budget elements. Only for the altimetry-based GrIS and AIS assessments, true month-to-month variability is not contained in the time series interpolated to monthly resolution. We found that the spread of monthly misclosure of OMB and of SLB is similar to, or narrower than, a Gaussian distribution with a standard deviation equal to the combined standard uncertainties of the budget elements.

## 7.2 Limitations of the study

While the uncertainty assessments made for the individual budget components were described in a common framework, different approaches to uncertainty characterisation were used for the different products. The reasons for the conceptual differences as well as their consequences for the relative uncertainty levels within the budget assessments have not been fully elaborated. A further consolidation and standardisation of uncertainty characterisation could allow, in a more flexible way, to propagate



uncertainties to different functionals, such as to anomalies with respect to different reference states, or
to time-dependent rates of change.
No correlation between errors of different budget elements were accounted for when combining the
different elements in budget assessments. However, such correlations exist. An important example is
the GIA correction, which is significant by its magnitudes and uncertainties. Errors in the GIA
corrections for GRACE OMC estimates (-1.37 ± 0.17 mm yr$^{-1}$) are likely correlated with errors of GIA
corrections for GRACE inferences of the ice-sheet contribution (-0.14 ± 0.09 mm yr$^{-1}$ for Antarctica; -
0.02 ± 0.02 mm yr$^{-1}$ for Greenland) and also correlated with the error of the GIA correction applied to
altimetric sea-level change (-0.30 ± 0.05 mm yr$^{-1}$).
As a matter of fact, already the choice of a GIA model used for GIA corrections poses consistency issues
not resolved in the present study. We used the regional GIA model IJ05_R2 (Ivins et al., 2013) for
GRACE-based AIS mass change estimates as opposed to the global GIA models used for GRACE-based
OMC estimates and GrIS mass change estimates. It is subject to ongoing controversy whether global
models are consistent with geodetic and geological evidence over Antarctica (Ivins et al., 2013; Argus
et al., 2014). Regional GIA models like IJ05_R2, W12 (Ivins et al., 2013; Whitehouse et al., 2012), on
the other hand, are not constructed to obey geological evidence on global sea-level history.
Contributions discussed in Sect. 3.8 that were not included in our budget analysis nor in our uncertainty
assessment are on the order of 0.1 mm yr$^{-1}$, with the largest single unconsidered contribution likely
being the elastic seafloor deformation effect.
It is also important to mention that our 'global' mean sea level assessment as well as our assessment of
the steric contribution, by its limitation to the 65°N–65°S latitude range, left out 6% of the global ocean
in the Arctic and in the Southern Ocean. In the polar oceans, satellite altimetry has sampling limitations
due to orbital geometry and sea-ice coverage. Likewise, Argo floats and other in-situ sensors have
sampling limitations due to the presence of sea ice. Therefore, SLB assessments for the polar oceans
(e.g., Raj et al., 2020) are even more challenging than for the 65°N–65°S latitude range focussed on in
this paper. An assessment of the truly global mean sea-level and its contributions would involve higher
uncertainties than quoted here for the 65°N–65°S range.

## 8 Data availability

A compiled dataset of time series of the elements of the GMSL budget and of the OMB together with
their        uncertainties,        are        freely        available        for        download        at
https://doi.org/10.5285/1562578dd07844f19f01f0db9366106d (Horwath et al., 2021). The single file in
csv format contains the time series presented in Fig. 10, 12 and 13. These time series are all at an
identical monthly sampling, resulting from interpolation of the original time series where necessary.
Uncertainties were partly recalculated from the original data products (as described in Sect. 3) in order
to make them consistently refer to anomalies with respect to the same reference interval Jan 2006 – Dec
2015 as stated in Sect. 2.3. Seasonal signals are removed from all time series.



# 9 Conclusions and outlook

This study assessed CCI data products related to the SLB, advanced the generation of new time series of SLB elements based on satellite earth observation and modelling, and integrated, within a consistent framework, the products into an analysis of the OMB and the SLB. The consolidation, improvement, and exhibition (in Fig. 1, 3–8) of the uncertainty characterization for every budget element were central to this study. The datasets and analyses presented here document both achievements and limitations identified within the SLBC_cci study.

## 9.1 Advances on data products on individual budget elements

For the GMSL, the use of the averaged ESA CCI 2.0 gridded sea-level data was enhanced by the incorporation of the uncertainty estimate over each GMSL time step from Ablain et al. (2019). Three major sources of errors were considered in the composition of a variance-covariance matrix to obtain GMSL uncertainty. The GMSL trend uncertainty over 1993–2016 (after correcting the TOPEX A drift) is assessed as 0.24 mm yr$^{-1}$ (1-sigma).

For the steric sea-level change, we developed a formal uncertainty framework around the estimation of steric height from Argo profiles, including propagation to gridded and time series products. The framework includes simple models to estimate each uncertainty source and their error covariance structures. Global sampling uncertainty was included when obtaining the global mean from the gridded products. Inclusion of SST from SST CCI to condition the climatology of the mixed layer reduced bias of the steric change in the upper ocean, with a small beneficial impact. A full error covariance matrix was calculated for the global steric time series, facilitating robust calculation of linear trends and their uncertainties.

OMC was inferred from recent GRACE SH solution releases. The choice of methodology built on comprehensive insights into the sensitivity to choices of input data and to choices of the treatment of background models.

For the glacier contribution, the introduction of an ensemble approach to reconstruct glacier mass change and the systematic multi-objective optimisation of the global model parameters led to results that generally confirm the previous estimates, and which also agree well with methods based on observations only (Zemp et al., 2019). However, the increased model performance (higher correlation with observations on individual glaciers, better representation of the observed variance of mass balance) increased the confidence in the results.

For the GrIS contribution, we devised an empirical and effective way to convert the radar altimetry elevation changes into mass changes. The resulting time series was independently tested against the GRACE-derived time series and it has shown very high compatibility.

For the AIS contribution, the new time series of Antarctica mass change from satellite radar altimetry is the result of an improved processing chain and a better characterisation of uncertainties. With a time-evolving ice and snow density mask and a new method for interpolating surface elevation change in areas located beyond the latitudinal limit of satellite radar altimeters and in between satellite tracks, we



have provided an updated time series of Antarctica mass change from 1992 to 2017 revealing that ice losses at Pine Island and Thwaites Glaciers basins are about 6 times greater than at the start of our survey.

For the LWS contribution, the version of the global hydrological model WaterGAP (version 2.2d) was developed and applied, which includes the commissioning years of individual reservoirs to take into account increased water storage behind dams as well as regionalised model parameterisations to improve the simulation of groundwater depletion (Müller Schmied et al., 2021). Comprehensive insights into the model sensitivity to choices of irrigation water use assumptions and climate input data were acquired, enabling a first uncertainty estimation. The good fit of simulated monthly total water storage anomaly (sum of land water storage and glacier storage) to GRACE-derived estimates, in particular regarding seasonality and de-seasonalised long-term variability, enhanced the confidence in the simulated land water contributions (Cáceres et al., 2020).

## 9.2 Sea-level budget and ocean-mass budget

As summarized in Table 3 and Table 4, the SLB and the OMB are closed within uncertainties for their evaluation periods P1 and P2 (SLB) and P2 (OMB). We may reformulate the budgets as follows. The GMSL linear trend over P1 and P2 is 3.05 mm yr$^{-1}$ and 3.64 mm yr$^{-1}$, respectively. The larger trend over P2 than over P1 is due to an increased mass component, predominantly from Greenland but also from the other mass contributors. Over P1 (P2) the steric contribution is 38% (33%) of GMSL rise, while the mass contribution is 57% (60%–66%). Among the sources of OMC, glaciers outside Greenland and Antarctica contributed 21% (21%) of total GMSL rise; Greenland contributed 20% (21%–24%); Antarctica contributed 6% (8%–9%); and LWS contributed 10% (11%). The SLB misclosure (GMSL minus sum of assessed contributions) is between +1% and +7% of GMSL rise. Ranges quoted here arise from different options of assessing the contributions. Uncertainties given in Table 3 and 4 are not repeated here.

We cannot attribute the statistically insignificant misclosure of linear trends. We tentatively attributed interannual features of misclosure to errors in some of the involved datasets. When the SLBC_cci steric product is used which uses constraints towards a static climatology, a SLB misclosure in the early years of Argo 2003–2006 is likely due to an underestimation of the steric sea-level rise. An interannual misclosure feature between 2006 and 2010 might be related to the SLBC_cci steric product and the WaterGAP model making the impression of a temporary slowdown in sea-level rise in 2006–2008 with subsequent recovery in 2008–2010, which is not as pronounced in the GMSL record.

## 9.3 Outlook

Future work will naturally include the extension of the considered time periods. It will be additionally spurred by the availability of new data types (Cazenave et al., 2019). GRACE-FO launched in August 2018 already facilitates a satellite gravity times series spanning 19 years (yet with interruptions). It will be equally important to continue this time series beyond GRACE-FO as currently jointly considered by ESA and NASA for the next generation gravity mission (Haagmans et al., 2020). The Deep Argo project



(Roemmich et al., 2019) promises new observational constraints on deep ocean steric contributions.
With the Sentinel-6/Jason-CS mission (Scharroo et al., 2016) the continuation of satellite altimetry in
the 66°N–66°S latitude range is enabled with synthetic aperture resolution capabilities exceeding those
of pulse-limited altimeters. Continuity of precise satellite radar altimetry at high latitudes beyond
CryoSat-2 still has to be ensured. Perspectives and requirements for long-term GMSL budget studies
are detailed by Cazenave et al. (2019). Additional ECVs related to the global water and energy cycle
call for their exploration in SLB studies. In the framework of ESA's CCI, results from Water Vapour
CCI, Snow CCI, or Lakes CCI are among the candidates.
Limitations discussed in Sect. 7.2 call for further methodological developments. For example, the
consideration of GIA as an own element in SLB analyses could help to enforce its consistent treatment.
This will be particularly important for regional SLB studies, since GIA is a driver of regional sea-level
change and OMC. Such a treatment of GIA could be in accord with the treatment of elastic solid-Earth
load deformations as proposed by Vishwakarma et al. (2020). Recent probabilistic characterizations of
GIA model errors (Caron et al., 2018) allow their propagation to error covariances of the SLB elements,
an approach not yet realized.
While GMSL is an important global indicator, it is indispensable to monitor and understand the
geographic patterns of sea-level change, that is, regional sea-level. Regional sea-level reflects the
different processes causing sea-level change, which may be hidden in GMSL (e.g., Stammer et al., 2013;
Hamlington et al., 2020). Understanding and projecting these processes, with implications down to
coastal impact research, is the ultimate goal. The further development of methodologies for regional
SLB assessments and their application will be an important step towards this goal.

## Author contributions

AC, JB, MH, AS, RF, CJM, JAJ, OBA, BM, FP, PD, KN designed the study. MH led the study and the
compilation and editing of the manuscript. AC, HKP, FM contributed the GMSL dataset and its
description. CJM, CRM, KvS contributed the SLBC_cci steric dataset and its description. BDG
contributed the GRACE-based OMC dataset and its description. BM contributed the global glacier
dataset and its description. VRB, RF contributed the GRACE-based Greenland dataset and its
description. LSS, SBS contributed the altimetry-based Greenland dataset and its description. AR, MH
contributed the GRACE-based Antarctica dataset and its description. AEH, AS, IO contributed the
altimetry-based Antarctica dataset and its description. PD, DC, HMS contributed the LWS dataset and
its description. BDG conducted the ocean mass budget analyses and drafted their description. HKP, AC,
MH conducted the sea level budget analyses and misclosure attribution study and drafted their
description. All authors discussed the results and contributed to the editing of the manuscript. KN
managed the project administration. JB launched the ESA CCI Sea Level Budget Closure Project
(SLBC_cci) and, with the support of MR, supervised the development of this research activity and
reviewed all the deliverables.



## Competing interests

The authors declare that they have no conflict of interest.

## Acknowledgements

This study was supported by the European Space Agency through the Climate Change Initiative project CCI Sea-level Budget Closure (contract number 4000119910/17/I-NB). We thank the German Space Operations Center (GSOC) of the German Aerospace Center (DLR) for providing continuously and nearly 100% of the raw telemetry data of the twin GRACE satellites.

## Appendix A

**Table A1:** Same as the last three columns of Table 4, but for the alternative period Jan 2005 – Aug 2016, when the Argo network was fully established. Linear trends of the sea-level budget elements [mm equivalent global mean sea-level per year].

| Budget element | Method | Jan 2005 – Aug 2016 | | |
|---|---|---|---|---|
| Total sea-level | altimetry | $3.80 \pm 0.28$ | $3.80 \pm 0.28$ | $3.80 \pm 0.28$ |
| Steric component | | | | |
| | SLBC_cci + deep steric estimate | $1.26 \pm 0.17$ | $1.26 \pm 0.17$ | $1.26 \pm 0.17$ |
| Glaciers | GGM | $0.78 \pm 0.04$ | $0.78 \pm 0.04$ | |
| Greenland | (altimetry) | $(0.72 \pm 0.07)$ | | |
| | (GGM) | $(0.20 \pm 0.03)$ | | |
| | Altimetry + GGM | $0.92 \pm 0.08$ | | |
| | GRACE | | $0.81 \pm 0.02$ | |
| Antarctica | altimetry | $0.42 \pm 0.04$ | | |
| | GRACE | | $0.31 \pm 0.11$ | |
| Land water storage | WaterGAP | $0.57 \pm 0.10$ | $0.57 \pm 0.10$ | |
| Sum of mass contributions | | $2.69 \pm 0.14$ | $2.47 \pm 0.15$ | |
| Ocean mass (65°N-65°S) | GRACE | | | $2.38 \pm 0.25$ |
| Sum of contributions | | $3.94 \pm 0.22$ | $3.73 \pm 0.23$ | $3.63 \pm 0.30$ |
| Misclosure | | $-0.14 \pm 0.36$ | $0.07 \pm 0.36$ | $0.17 \pm 0.41$ |

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
