# Peer review of "Global sea-level budget and ocean-mass budget, with focus on advanced data"

_Earth System Science Data, 2021_

## Referee Comment (RC2)

Formal Review of ESSD-2021-137 (Horwath et al., 2021)

This manuscript provides a fairly comprehensive analysis of global sea level and ocean mass budget and uncertainty characterization. It is well written and organized, and my comments are mostly minor and can help the authors to improve the presentation. There are some issues the authors need to clarify.

Comments:

1) In the abstract, the authors discussed the SLB and OMB analyses for two periods (P1 & P2). However, for P2 only the mass term was discussed here. The steric component should be discussed as well.

2) Line 39: " … sea-level change and its contributions." "Contributors" appears to be the right word to use here.

3) Lines 108-110: References are needed to support the conclusions. "GRACE analyses suggest LWS gains and therefore a negative GMSL contribution from LWS" – I believe this was based on only one previous study (Reager et al., 2016). Some other later studies came to a completely different conclusion. In addition to the two (Cáceres et al., 2020; Gutknecht et al., 2020) cited later in Discussion by the authors, I would suggest to also cite Kim et al. (2019), the first to have reported a different and opposite GRACE LWS contribution to GMSL. A brief discussion of the different LWS estimates is needed here.

   *Kim, J.-S., Seo, K.-W., Jeon, T., Chen, J., & Wilson, C. R. (2019). Missing hydrological contribution to sea level rise. Geophysical Research Letters, 46. https://doi.org/10.1029/2019GL085470*

4) Line 144: "other" should be added to the list, consistent with other occurrences.

5) Line 174: "The SLB The presentation of the result needs substantial improvements. The authors should at least provide a clear definition of the two studied drainage basins, and show how the defined GRACE mascons are configured in the basins. The two maps in Figure 1 are simply not sufficient. The authors should also provide a plot showing GRACE-derived ice mass change time series for the two basins.

6) Line 226: Please clarify what interpolation method(s) is (are) used here.

7) Lines 267-268: CMEMS provides altimeter sea level anomaly grids for the entire altimeter period, the authors need to explain why they decided to combine the two datasets (CCI sea-level record & CMEMS) to get the GMSL series and not use the CMEMS series for consistency.

8) Lines 365-366: "It is relatively common for there to be layers with no observations, sometimes in the upper ocean and often at depth." Please rephrase this sentence. It doesn't sound grammatically correct.

9) Lines 494-498: The authors used CSR, JPL and GFZ GRACE RL06 SH gravity solutions (together with ITSG-Grace2018), but decided to use old generations of geocenter (Swenson et al., 2008) and SLR C20 series (GRACE TN-11), which were prepared for GRACE RL05 solutions. Are there any logical reasons for not using the RL06 supplementary datasets? Using RL05 SLR C20 may be fine, but the new RL06 geocenter

series are expected to have substantially different effects on GRACE ocean mass rate estimates.

10) Lines 499-509: Please clarify which ICE6G GIA model was used. The authors cited Peltier et al. (2015), which is an outdated version. After fixing some error(s), Peltier et al. (2018) released an updated version of the model, ICE6G_D.

As seen from the two comments above (9 & 10), the GRACE related analyses in the current study are not up to the current standards. The authors need to either provide a convincing reasoning or update the analyses using the current standards.

11) Figure 10: The authors may choose some colors to better distinguish the curves (red and magenta are not a good pair).

12) Line 1144: "We cannot attribute the misclosures in the budgets of linear trends, …" Please consider rephrasing this sentence. "We cannot attribute the misclosures in the budgets of linear trends to any particular error source, …" ?

13) Lines 115-1156: "The unassessed atmospheric water content contribution (cf. Sect. 3.8) could add to the misclosure, though." Please consider rephrasing this sentence.

"The unassessed atmospheric water content contribution (cf. Sect. 3.8) could contribute to the misclosure, though."?

14) Line 1182: "are0.26" -> "are 0.26"

15) Lines 1216-1220: "Errors in …" It is unclear what are really discussed here. Do those cited values (e.g., $-1.37 \pm 0.17$ mm yr$^{-1}$) represent estimates rates with uncertainties? If so, the values for Antarctic ($-0.14 \pm 0.09$) and Greenland ($0.02 \pm 0.02$) do not sound right.

16) Lines 1284-1287: "we have provided …" It is not clear what the "start of our survey" is referred to. Please be more specific.

Jianli Chen

---

## Author Comment (AC2)

**Authors' response to reviewer 1**

We thank the reviewer for the constructive and insightful comments which helped us to improve the manuscript.

In the following we respond to the specific comments. (Reviewer comments are repeated in *italic.*)

*1) Part 9.2 line 1312. Suggestions regarding " a SLB misclosure in the early years of Argo 2003–2006 is likely due to an underestimation of the steric sea-level rise". I would suggest to include the reference to support the statement regarding the underestimation of the steric sea-level rise, or add a sentence to clarify this statement.*

We elaborated this statement and made explicit reference to Section 3.2.2 where it is discussed in detail. Now the sentence reads: "When the SLBC_cci steric product is used, a SLB misclosure in the early years of Argo 2003–2006 is likely due to an underestimation of the steric sea-level rise associated to the global sampling error in conjunction with the constraints towards a static climatology, as discussed in Sect. 3.2.2."

*2) Part 3.2.3 Deep ocean steric contribution*
*This part needs some comment (just a comment) regarding the estimate… 0.1 ± 0.1 mm/yr based on Purkey and Johnson (2010), to address the optimistic uncertainties in this estimate, suggesting that some very limited data sets are available to calculate this estimate, mainly to bring the issue of the need for observations in the deep ocean.*

We added two sentences to discuss the issue: "Note that this estimate is based on sparse in situ sampling. Corresponding evolutions of the ocean observing system are under way (Roemmich et al., 2019)."

*3) It would be good to see some statement, providing a clear message (maybe in abstract), that availability of the data (e.g. results from P1 vs P2) is important to improve our understanding about sea level components, and to understand the source of uncertainties…. Something like that. However, I do not inflict any specific statement, I just would like to suggest a message about the importance of observations for sea level components. It is a very minor suggestion.*

Thank you for this suggestion. We added a sentence in the Introduction, §2: "Clearly, as a prerequisite of progress in SLB studies, datasets on the mentioned budget elements must be accessible."

---

## Author Comment (AC3)

**Authors' response to reviewer 2 (Jianli Chen)**

We thank the reviewer for the constructive and insightful comments which helped us to improve the manuscript. Following reviewer comments 9 and 10 we also updated our analysis for GRACE ocean mass change (see below).

In the following we respond to the specific comments. (Reviewer comments are repeated in *italic*.)

1) In the abstract, the authors discussed the SLB and OMB analyses for two periods (P1 & P2). However, for P2 only the mass term was discussed here. The steric component should be discussed as well.

We added the numbers for the steric contribution for period P2.

2) Line 39: "... sea-level change and its contributions." "Contributors" appears to be the right word to use here.

We replaced "contributions" by "contributors".

3) Lines 108-110: References are needed to support the conclusions. "GRACE analyses suggest LWS gains and therefore a negative GMSL contribution from LWS" – I believe this was based on only one previous study (Reager et al., 2016). Some other later studies came to a completely different conclusion. In addition to the two (Cáceres et al., 2020; Gutknecht et al., 2020) cited later in Discussion by the authors, I would suggest to also cite

*Kim et al. (2019), the first to have reported a different and opposite GRACE LWS contribution to GMSL. A brief discussion of the different LWS estimates is needed here.*

Kim, J.-S., Seo, K.-W., Jeon, T., Chen, J., & Wilson, C. R. (2019). Missing hydrological contribution to sea level rise. Geophysical Research Letters, 46. https://doi.org/10.1029/2019GL085470

Thank you for pointing at our omission of relevant work here. We draw a more complete picture now by writing: "Determining the LWS contribution to sea-level is a particular challenge (WCRP, 2018): Hydrological models generally suggest LWS losses and therefore a positive contribution from LWS to GMSL rise (Dieng et al., 2017; Scanlon et al., 2018; Cáceres et al., 2020). Initial GRACE-based estimates indicated a gain of LWS (Reager et al. 2016; Rietbroek et al. 2016), while newer GRACE-based estimates (Kim et al. 2019, Frederikse et al. 2020) agree with global hydrological modeling results on the sign of change (loss of LWS). Moreover, in view of the high interannual variability of LWS, the determined trend strongly depends on the selected time period and method of trend determination."

4) Line 144: "other" should be added to the list, consistent with other occurrences.

We added "or other sources" in the sentence after Eq. 6.

5) Line 174: "The SLB The presentation of the result needs substantial improvements. The authors should at least provide a clear definition of the two studied drainage basins, and show how the defined GRACE mascons are configured in the basins. The two maps in Figure 1 are simply not sufficient. The authors should also provide a plot showing GRACE-derived ice mass change time series for the two basins.

In the meantime, it was clarified with the reviewer that comment 5 slipped into the review by mistake, as it did not refer to our manuscript.

6) Line 226: Please clarify what interpolation method(s) is (are) used here.

We now specified that the interpolation was linear.

7) Lines 267-268: CMEMS provides altimeter sea level anomaly grids for the entire altimeter period, the authors need to explain why they decided to combine the two datasets (CCI sealevel record & CMEMS) to get the GMSL series and not use the CMEMS series for consistency.

The principle of our study was to use CCI products wherever available. We now made this clear by writing: "While our study focusses on utilising CCI products, the CCI sea-level product did not cover the year 2016. We

therefore extended the GMSL record with the Copernicus Marine Environment and Monitoring Service dataset (CMEMS, https://marine.copernicus.eu/) from Jan 2016 to Dec 2016."

8) Lines 365-366: "It is relatively common for there to be layers with no observations, sometimes in the upper ocean and often at depth." Please rephrase this sentence. It doesn't sound grammatically correct.

We made the sentence simpler. It now reads: "It is relatively common to have layers ..."

9) Lines 494-498: The authors used CSR, JPL and GFZ GRACE RL06 SH gravity solutions (together with ITSG-Grace2018), but decided to use old generations of geocenter (Swenson et al., 2008) and SLR C20 series (GRACE TN-11), which were prepared for GRACE RL05 solutions. Are there any logical reasons for not using the RL06

supplementary datasets? Using RL05 SLR C20 may be fine, but the new RL06 geocenter series are expected to have substantially different effects on GRACE ocean mass rate estimates.

See our response to comment 10

10) Lines 499-509: Please clarify which ICE6G GIA model was used. The authors cited Peltier et al. (2015), which is an outdated version. After fixing some error(s), Peltier et al. (2018) released an updated version of the model, ICE6G\_D.

As seen from the two comments above (9 & 10), the GRACE related analyses in the current study are not up to the current standards. The authors need to either provide a convincing reasoning or update the analyses using the current standards.

We agree that the standards on degree-one time-series and some other aspects of GRACE analysis have changed since the time when our original GRACE-based OMC products were generated.

Given the pertinent comments by two reviewers (Reviewer 2 comments #9 and #10; Reviewer 3 comment on line 495) we decided to perform a major update of the GRACE-based OMC analysis according to more recent standards. The update concerns GIA corrections, Degree-one solutions, and C20 series. The update includes updates of the related uncertainty assessment. The update has resulted in an increase of the linear trends for GRACE ocean mass change in the order of several tenths of millimetres per year. The updated standard uncertainties are also increased slightly.

This update entailed a complete re-work of the budget analysis and a change to many numbers. We believe that it was worth the effort and that the dataset and assessment thus provided may serve the community as a longer-lasting reference.

Some more details on the update are appended at the end of this reply.

We have updated the pertinent text in Sect. 3.3 as well as all numbers and figures in Sections 4, 5, 7, and 8 that depend on the OMC products.

11) Figure 10: The authors may choose some colors to better distinguish the curves (red and magenta are not a good pair).

We have changed the colors. They are now dark red and light red and should be better distinguishable than previously

12) Line 1144: "We cannot attribute the misclosures in the budgets of linear trends, ..." Please consider rephrasing this sentence. "We cannot attribute the misclosures in the budgets of linear trends to any particular error source, ..."?

We rephrased the sentence accordingly.

13) Lines 115-1156: "The unassessed atmospheric water content contribution (cf. Sect. 3.8) could add to the misclosure, though." Please consider rephrasing this sentence. "The unassessed atmospheric water content contribution (cf. Sect. 3.8) could contribute to the misclosure, though."?

Well observed. We rephrased the sentence accordingly.

14) Line 1182: "are0.26" -> "are 0.26"

Thanks for spotting this. We added the space

15) Lines 1216-1220: "Errors in ..." It is unclear what are really discussed here. Do those cited values (e.g., -  $1.37 \pm 0.17$  mm yr1) represent estimates rates with uncertainties? If so, the values for Antarctic (-0.14 ± 0.09) and Greenland (0.02 ± 0.02) do not sound right.

We reworded this part and put it into two separate sentences. They should now be clearer. We also checked the numbers. The part now reads:

In our study, the GIA corrections and their uncertainties are  $-1.37 \pm 0.19$  mm yr-1 for the GRACE OMC estimate, -0.14 ± 0.09 mm yr-1 and -0.02 ± 0.02 mm yr-1 for the GRACE-based assessment of the Antarctic and Greenland mass contribution, respectively, and -0.30 ± 0.05 mm yr-1 for the altimetric GMSL change. (These numbers are subtracted from the uncorrected results.) The errors in these GIA corrections to different budget elements are likely correlated among each other.

16) Lines 1284-1287: "we have provided ..." It is not clear what the "start of our survey" is referred to. Please be more specific.

Thanks four pointing on this. We rephrased the sentence. It now reads:

This new dataset (cf. Shepherd et al., 2019) shows that ice losses are dominated by the Pine Island Glacier and Thwaites Glacier basins in West Antarctica, where mass losses (expressed as equivalent GMSL contribution) have increased from  $0.04 \pm 0.01$  mm yr-1 in 1992–1997 to  $0.36 \pm 0.03$  mm yr-1 in 2012–2017.

**Additional background information on the GRACE ocean mass change analysis update**

The following information is given as part of the response to the reviewers, in addition to the information given in the revised manuscript.

**Approach to assessment of trend uncertainty associated to degree-one, C20, and GIA**

Trend uncertainties associated go degree-one, C20, and GIA were assessed individually based on the spread of a small ensemble of different options to incorporate these effects.

For example, for degree-one, ten different degree-one time series were used. There temporal sampling was identical to that of the GRACE solutions. They were treated in the same way as the GRACE solutions (synthesis into the spatial domain, weighted integration with a buffered ocean kernel, scaling to account for the buffer). In this way, time series of the degree-one contribution to the OMC estimate were derived. Linear trends of these time series were calculated according to Section 2.2. The standard deviation of the ten linear trend values was calculated and taken as the standard error associated to Degree-one. We did not adjust the calculation of the standard deviation to effects of the low sample size (e.g., with t-Student distributions). The same procedure was used for C20. For GIA, the procedure was performed immediately on the level trends, rather than by generating time series and fitting trends to them.

Differences between GIA models affect differences between GIA corrections as well as differences between degree-one solutions. The former were addressed by the GIA uncertainty assessment, while the latter were included in the degree-one uncertainty assessment.

**Degree-one**

We agree that the formerly used geocenter time series are outdated from today's perspective. The choice had been a question of project timelines (involving "freezing" of data products) rather than of awareness. For our update, we skip the former single degree-1 solution and use coefficients according to more recent standards, which indeed affects trends significantly. However, we do not simply take the TN13 products (which basically is Sun et al. (2016) with ICE-6G\_D GIA), but we compute our own in-house solution according to Sun's method. We do this because, first, TN13 does not provide an individual degree-one product for ITSG-Grace2018, and second, it allows us to be consistent with varying GIA choices. It means we have a dedicated degree-one correction for each combination of GRACE solution {ITSG, CSR, GFZ, JPL} and GIA model {A2013, Caron, Peltier}.

As expected, using these degree-one solutions instead of those in the previous manuscript version results in a notable increase of OMC trends in the order of tenths of a millimetres per year. Now all tested combinations result in a linear global ocean mass trend of more than 2 millimetres per year over the period P2.

We reran the degree-one uncertainty assessment and found an increase of the 1-sigma standard deviation of trends from 0.14 mm/a (old) to 0.18 mm/a (new). This time we picked a variety of ten different solutions out of a total pool of twenty options and tried to keep a balance between age, choice of method, data centres and GIA models:

- Swenson 08, CSR RL05, Tellus (A2013)
- Rietbroek 2016 (combination approach)
- Cheng 2010 SLR
- Sun 2016, CSR RL06, orig. (A2013)
- 3 x TN13 (CSR, GFZ, JPL), with ICE-6G\_D
- 3 x TUDresden GEF, ITSG-Grace2018, GIA from {A2013, Caron18, ICE-6G\_D}

Figure 1 compares our degree-one time-series (when employing CSR RL06 and ICE-6G\_D) to the degree-one time-series provided at Tellus for CSR RL05 and CSR RL06. Figure 2 shows the degree-one contributions to the

global OMC estimate according to the ten solutions that underlie our uncertainty assessment. Figure 3 shows the distribution of associated linear trends.

Figure 1: Degree-one time-series expressed in terms of geocenter motion [mm]. Comparison between Tellus RL05, Tellus RL06 and our dedicated solution (red) based on an approach comparable to the TN-13 approach (based on Sun et al., 2016); in this case for the combination of GRACE RL06 CSR with ICE-6G\_D.

---

## Author Comment (AC4)

**Authors' response to reviewer 3 (Riccardo Riva)**

We thank the reviewer for the constructive and insightful comments which helped us to improve the manuscript. Following the reviewer's comment on line 495 we also updated our analysis for GRACE ocean mass change (see below).

In the following we respond to the specific comments. (Reviewer comments are repeated in *italic.*)

*Line 93: in the list of recent studies, I miss Frederikse et al. (Nature 584, 2020). Note that their numbers (Table 1) are very similar to those presented in the current manuscript.*

We included reference to Frederikse et al. (2000). In Section 7.1 we discussed the good agreement between Frederikse et al. (2000) and our study.

*Line 158: though it is common practice, I am still not convinced that it is correct to use the density of freshwater when computing sea level change induced by continental freshwater fluxes. Freshwater will probably mix rather quickly (surely when looking at global values over long times) and salt content will cause a reduction in water volume. It is a somehow small effect, hence below the uncertainty level, but it could at least be mentioned either here or in the later discussion of unaccounted error sources.*

We mentioned the issue of different water densities used in the literature in Section 3.8. We now refer to this discussion right at the introduction of $\rho_W$ at the place addressed by the reviewer.

*Line 190: the explanation of the use of "GlobalOcean" vs. "Ocean" is rather unclear.*

We agree that this formulation was unnecessarily complicated. We modified the notation and its explanation. It now reads:

"Based on such assumptions, $\Delta SL_{source}$ may be evaluated as

$$\Delta SL_{source} = \frac{1}{\rho_W} \langle \Delta \kappa_{source} \rangle_{Ocean65}, \qquad\qquad (8)$$

where $\langle \cdot \rangle_{Ocean65}$ denotes the averaging over the ocean area between 65°N and 65°S. Here we assume $\langle \Delta \kappa_{source} \rangle_{Ocean65} = \langle \Delta \kappa_{source} \rangle_{Ocean}$."

*Line 279: it would be nice to show the mentioned trend of 3.05 ± 0.24 mm/yr in Figure 1a (both the straight line and the shaded uncertainty). It would help visualizing the presence of non-linear changes.*

Our idea is to present the time series and time-variable uncertainties for all budget elements in the same style (Fig. 1, 2, 4, 5, 6, 7, 8). Partly, it would complicate readability if we added the line / or lines of fitted models (potentially different lines for different time intervals). Therefore, we suggest to refrain from adding this line and to concentrate on showing the actual data.

*Line 304: when mentioning the 1.65-sigma uncertainty, I would also add that it is equivalent to the 90% confidence margin. It might not be obvious to everybody.*

Good point. We added an according phrase

*Line 365: "for there to be …" could be changed into "for … to be present".*

We made the sentence simpler. It now reads: "It is relatively common to have layers …"

*Line 482: when I first read about the use of a scaling factor, I was worried it would introduce unknown biases. Only later on (lines 555 and 579), I was convinced that it is indeed appropriate to make use of such a*

*strategy. I suggest adding a comment, possibly with a caveat and reference to the appropriate section, that the effect of using of a scaling factor has been explicitly analysed.*

We made this point clear by referring to the underlying assumption right at the place which used to be line 482. It now reads: "This scaling is based on the assumption that the mean EWH change in the buffer equals the mean EWH change in the buffered ocean area. Effects of violations to this assumption are included in the uncertainty assessment (see further below)."

*Line 495: it is somehow surprising that the authors did not make use of the most recent degree-one timeseries provided by NASA-JPL, as they did for C20. It requires some motivation for this choice. I would also recommend to perform a comparison about the timeseries used in this study and those available through podaac-tools.jpl.nasa.gov.*

We agree that the standards on degree-one time-series and some other aspects of GRACE analysis have changed since the time when our GRACE-based OMC products were generated.

Given the pertinent comments by two reviewers (Reviewer 2 comments #9 and #10; Reviewer 3 comment on line 495) we decided to perform a major update of the GRACE-based OMC analysis according to more recent standards. The update concerns GIA corrections, Degree-one solutions, and C20 series. The update includes updates of the related uncertainty assessment. The update has resulted in an increase of the linear trends for GRACE Ocean Mass Change in the order of several tenths of millimetres per year. The updated standard uncertainties are also increased slightly.

This update entailed a complete re-work of the budget analysis and a change to many numbers. We believe that it was worth the effort and that the dataset and assessment thus provided may serve the community as a longer-lasting reference.

Some more details on the update are appended at the end of our reply to Reviewer 1. They include comparisons among many variants of handling GIA, degree-one and C20.

We have updated the pertinent text in Sect. 3.3 as well as all numbers and figures in Sections 4, 5, 7, and 8 that depend on the OMC products.

*Line 577: it is unclear how degree-one and C20 uncertainties have been determined, considering that only a single product was used. "The same approach" seems to refer to the GIA uncertainty, which is based on three different models.*

C20 uncertainties and degree-one uncertainties were assessed based on ensembles, too. We added about 10 lines of text to explain this assessment in more detail.

Even more background information is appended at the end of our response to Reviewer 2.

*Line 664: "Figure 5c" should be "Figure 5b".*

Thanks for spotting this typo. We corrected it.

*Line 732: the reference to Simonsen et al (2021) could be removed, since it is already mentioned two lines later (or the fact that the approach follows Simonsen could be mentioned earlier).*

We shifted the sentence "This approach follows that of Simonsen et al., 2021." To the place where the description of the approach starts. We deleted the second, redundant reference to Simonsen et al., 2021

*Line 736: what approach was used to exclude the peripheral glaciers from the grid?*

We added more detail and a reference. It now reads: "The peripheral glaciers (connectivity level 0 and 1 according to Rastner et al. 2012) were excluded from the grid.

*"Line 751: probably "reduced" (or something similar) is more appropriate than "circumvented".*

Yes. We replaced "circumvented" by "reduced"

*Line 801: I find it a bit of a pity that Ivins et al. (2013) has been used here instead of Caron et al. (2018) as elsewhere in the manuscript. Admittedly, in the discussion section the inconsistent treatment of GIA is explicitly mentioned as a limitation of this study. I can imagine that there were practical reasons for this choice, but it does require a short motivation.*

The practical reason is that the study used CCI products where possible. Therefore, we used the AIS CCI Gravimetric Mass Balance products which use the Ivins et al. (2016) GIA model. We made this clearer by changing the wording. The question which GIA model is closest to the truth for Antarctica is open, of course. We now explicitly refer to the pertinent discussion in Section 7.2. So this part in Section 3.6.1 now reads: "The GIA correction adopted by these products was based on the regional model by Ivins et al. (2013). In Sect. 7.2 we address the trade-off between using global or regional GIA models for Antarctica."

*Line 850: the whole paragraph about Figure 7a would better fit after line 824, before the uncertainty assessment.*

We followed the suggestion and shifted the discussion of results to the place before the uncertainty assessment. We reworded some sentences to make them better flow in their new context.

*Line 939: it is nice that the other contributions are explicitly discussed, but it is unclear why they have not been added to the final budget.*

We agree that the study is not conclusive about the contributions and issues discussed in Section 3.8. Previous assessments of these contributions and issues (shortly reviewed in Sect. 3.8) are partly in conflict, refer to different time intervals, and do not always comprise time series.

Our principle has been to exercise our analysis based on datasets to which we have thorough insights, as outlined in Section 1. (Admittedly, we were not puristic about this principle when adopting a deep ocean steric estimate.) Therefore, we suggest to leave a more comprehensive inclusion of the mentioned contributions to future work.

Back-of-the-envelope calculations indicate that accounting for the missing contributions would not change the conclusions of the study on budget closure. We added this as a comment in the discussion section.

*Table 3: please add a label, and a short explanation in the caption, about the difference between the two rightmost columns. Since some components are used in both columns, their meaning is not evident.*
*Table 4: same comment as for Table 3, this time about the three rightmost columns.*

Thanks for pointing to this potential source of confusion. We largely extended the Table captions for Table 3 and 4 and added footnotes to explain the meaning of the entries in the different columns.

*Line 1040: the proof of the Gaussian error distribution assumption is very nice, maybe a comment about it could be anticipated in the Methods section.*

Thank you, this is a good hint. We added a pertinent explanation after Eq. (7) in Section 2.1

*Line 1121: the fact that percentage of misclosures does not follow a Gaussian distribution is worth a comment.*

We added a comment that the distribution is again (as in the previous paragraph) narrower than allowed for by the assessed uncertainties. We do no elaborate on whether the distribution is non-Gaussian or just Gaussian with a smaller width.

*Line 1141: same as previous comment.*

Again, we added a respective comment.

*Line 1198: again, the same numbers for the LWS contribution also found by Frederikse et al. (2020).*

Yes. We now added reference to, and comparison with, Frederikse et al. (2000) in Section 7.1

*Line 1228: same comment as about line 939.*

See our response to the comment about line 939. As a compromise, we added a sentence here saying: "Coarse estimates based on the literature review of Sect. 3.8 indicate that considering the discussed effects does not change the overall conclusions of our study."

*Line 1237: the fact that the budget excludes polar areas is explained in the beginning of the manuscript, but it could also be repeated in the Table captions. This because some people will pull and use numbers from the tables without actually reading the full manuscript.*

The original manuscript already stated in the Caption of Table 4: "The estimates of total sea level, the steric contribution, and the GRACE-based OMC refer to the ocean between 65°N and 65°S". We now added: "thereby excluding polar and subpolar oceans in the Arctic and the Southern Ocean."

*Line 1239: the section about data availability would make more sense after the conclusions.*

The manuscript composition suggested by the ESSD journal (https://www.earth-system-science-data.net/submission.html) has the "Data availability" section prior to the "Conclusions" section, thus qualifying the "Data availability" as being a generic part of the manuscript rather than a kind of addendum. We suggest to keep this order.

---

## Author Response (AR2)

Dear Dr. Manuella,

thank you for your support with this manuscript.

We changed the text on line 592ff according to your recommendation not to refer to the ESSD Discussions manuscript.

We shortened the text and referred to a different source (Horwath et al. 2019) to specify the previous version of our assessment, to which the current version is an update.

Best regards,

Martin Horwath